# Internal models for interpreting neural population activity during sensorimotor control

Matthew D Golub[1,2], Byron M Yu[1,2,3*†], Steven M Chase[2,3*†]

[1]Department of Electrical and Computer Engineering, Carnegie Mellon University, Pittsburgh, United States; [2]Center for the Neural Basis of Cognition, Carnegie Mellon University, Pittsburgh, United States; [3]Department of Biomedical Engineering, Carnegie Mellon University, Pittsburgh, United States

**Abstract** To successfully guide limb movements, the brain takes in sensory information about the limb, internally tracks the state of the limb, and produces appropriate motor commands. It is widely believed that this process uses an internal model, which describes our prior beliefs about how the limb responds to motor commands. Here, we leveraged a brain-machine interface (BMI) paradigm in rhesus monkeys and novel statistical analyses of neural population activity to gain insight into moment-by-moment internal model computations. We discovered that a mismatch between subjects' internal models and the actual BMI explains roughly 65% of movement errors, as well as long-standing deficiencies in BMI speed control. We then used the internal models to characterize how the neural population activity changes during BMI learning. More broadly, this work provides an approach for interpreting neural population activity in the context of how prior beliefs guide the transformation of sensory input to motor output.

**\*For correspondence:** byronyu@cmu.edu (BMY); schase@cmu.edu (SMC)

[†]These authors contributed equally to this work

**Competing interests:** The authors declare that no competing interests exist.

## Introduction

Even simple movements, like reaching to grasp a glass of water, require dozens of muscles to be activated with precise coordination. This precision is especially impressive in light of sensory feedback delays inherent to neural transmission and processing: when we make a swift arm movement, the brain only knows where the arm was a split second ago, not where it currently is. To generate the desired movement, it is widely believed that we form internal models that enable selection of appropriate motor commands and prediction of the outcomes of motor commands before sensory feedback becomes available (*Crapse and Sommer, 2008*; *Shadmehr et al., 2010*).

Mechanistic studies have made important progress toward identifying the neural circuits that implement internal models in sensory (*Komatsu, 2006*; *Kennedy et al., 2014*; *Schneider et al., 2014*), vestibular (*Laurens et al., 2013*), and motor (*Sommer, 2002*; *Ghasia et al., 2008*; *Keller and Hahnloser, 2009*; *Azim et al., 2014*) systems. In parallel, psychophysical studies have demonstrated the behavioral correlates of these internal models (*Shadmehr and Mussa-Ivaldi, 1994*; *Wolpert et al., 1995*; *Thoroughman and Shadmehr, 2000*; *Kluzik et al., 2008*; *Mischiati et al., 2015*) and the behavioral deficits that result from lesions to corresponding brain areas (*Shadmehr and Krakauer, 2008*; *Bhanpuri et al., 2013*). Together with studies showing neural correlates of internal models (*Sommer, 2002*; *Gribble and Scott, 2002*; *Ghasia et al., 2008*; *Mulliken et al., 2008*; *Keller and Hahnloser, 2009*; *Green and Angelaki, 2010*; *Berkes et al., 2011*; *Laurens et al., 2013*), these previous studies have provided strong evidence for the brain's use of internal models.

**eLife digest** The human brain is widely hypothesized to construct "inner beliefs" about how the world works. It is thought that we need this conception to coordinate our movements and anticipate rapid events that go on around us. A driver, for example, needs to predict how the car should behave in response to every turn of the steering wheel and every tap on the brake. But on icy roads, these predictions will often not reflect how the car would behave. Applying the brakes sharply in these conditions could send the car skidding uncontrollably rather than stopping. In general, a mismatch between one's inner beliefs and reality is thought to cause errors and accidents. Yet this compelling hypothesis has not yet been fully investigated.

Golub et al. investigated this hypothesis by conducting a "brain-machine interface" experiment. In this experiment, neural signals from the brains of two rhesus macaques were recorded using arrays of electrodes and translated into movements of a cursor on a computer screen. The monkeys were then trained to mentally move the cursor to hit targets on the screen.

The monkeys' cursor movements were remarkably precise. In fact, the experiment showed that the monkeys could internally predict their cursor movements just as a driver predicts how a car will move when turning the steering wheel. These findings indicate that the monkeys have likely developed inner beliefs to predict how their neural signals drive the cursor, and that these beliefs helped coordinate their performance.

In addition, when the monkeys did make mistakes, their neural signals were not entirely wrong—in fact they were typically consistent with the monkeys' inner beliefs about how the cursor moves. A mismatch between these inner beliefs and reality explained most of the monkeys' mistakes.

The brain constructs such inner beliefs over time through experience and learning. To study this learning process, Golub et al. next conducted an experiment in which the cursor moved in a way that was substantially different from the monkey's inner beliefs. This experiment uncovered that, during the course of learning, the monkey's inner beliefs realigned to better match the movements of the new cursor. Taken together, this work provides a framework for understanding how the brain transforms sensory information into instructions for movement. The findings could also help improve the performance of brain-machine interfaces and suggest how we can learn new skills more rapidly and proficiently in everyday life.

These internal models are presumably rich entities that reflect the multi-dimensional neural processes observed in many brain areas (*Cunningham and Yu, 2014*) and can drive moment-by-moment decisions and motor output. However, to date, most studies have viewed internal models through the lens of individual neurons or low-dimensional behavioral measurements, which provides a limited view of these multi-dimensional neural processes (although see *Berkes et al., 2011*). Here, we address these limitations by extracting a rich internal model from the activity of tens of neurons recorded simultaneously. The key question that we ask is whether such an internal model can explain behavioral errors that cannot be explained by analyzing low-dimensional behavioral measurements in isolation.

We define an internal model to be one's inner conception of a motor effector, which includes one's prior beliefs about the physics of the effector as well as how neural commands drive movements of the effector. When we extract a subject's internal model, we seek a statistical model of the effector dynamics that is most consistent with the subject's neural commands. Interpreting high-dimensional neural activity through the lens of such an internal model offers insight into how one's prior beliefs about the effector affect the transformation of sensory inputs into population-level motor commands on a timescale of tens of milliseconds.

To date, it has been difficult to identify such an internal model due the complexities of non-linear effector dynamics and multiple sensory feedback modalities, the need to monitor many neurons simultaneously, and the lack of an appropriate statistical algorithm. To overcome these difficulties, we leveraged a closed-loop brain-machine interface (BMI) paradigm (*Figure 1A*) in rhesus monkeys, which translates neural activity from the primary motor cortex (M1) into movements of a computer cursor (*Green and Kalaska, 2011*). A BMI represents a simplified and well-defined feedback control

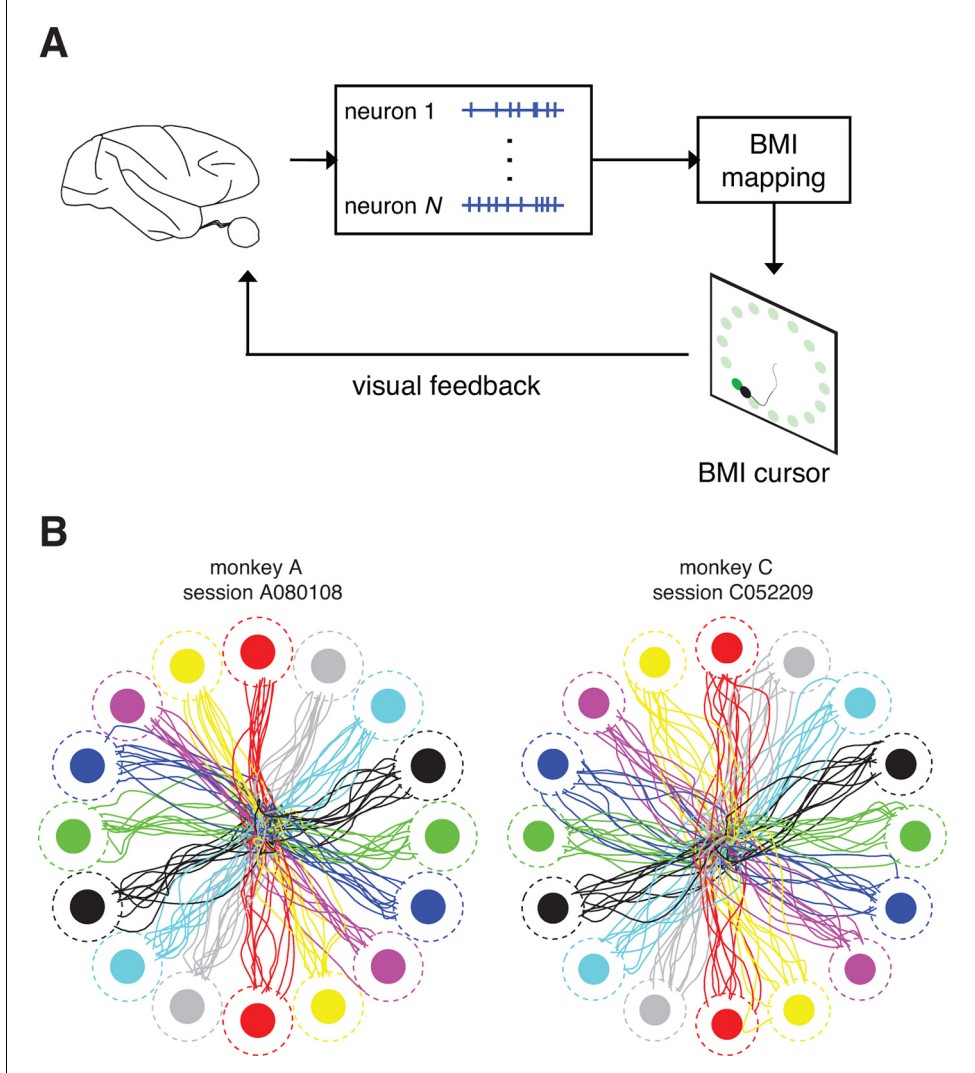

**Figure 1.** Closed-loop control of a brain-machine interface (BMI) cursor. (**A**) Schematic view of the brain-machine interface. Subjects produce neural commands to drive a cursor to hit visual targets under visual feedback. (**B**) Cursor trajectories from the first 10 successful trials to each of 16 instructed targets (filled circles) in representative data sets. Target acquisition was initiated when the cursor visibly overlapped the target, or equivalently when the cursor center entered the cursor-target acceptance zone (dashed circles). Trajectories shown begin at the workspace center and proceed until target acquisition. Data are not shown during target holds.
The following figure supplement is available for figure 1:

**Figure supplement 1.** Proficient control of the brain-machine interface (BMI).

system, which facilitates the study of internal models (*Golub et al., 2016*). In particular, the BMI mapping from neural activity to movements is completely specified by the experimenter and can be chosen to define linear cursor dynamics, the relevant sensory feedback can be limited to one modality (in this case, vision), and all neural activity that directly drives the cursor is recorded.

During proficient BMI control, as with other behavioral tasks, subjects make movement errors from time to time. One possible explanation for these errors is that they arise due to sensory or motor "noise" that varies randomly from one trial to the next (*Harris and Wolpert, 1998*; *Osborne et al., 2005*; *Faisal et al., 2008*). Another possibility, which is the central hypothesis in this study, is that a substantial component of movement errors is structured and can be explained by a mismatch between the subject's internal model of the BMI and the actual BMI mapping.

Testing this hypothesis required the development of a novel statistical method for estimating the subject's internal model from the recorded M1 activity, BMI cursor movements, and behavioral task goals. The internal model represents the subject's prior beliefs about the physics of the BMI cursor, as well as how the subject's neural activity drives the cursor. To justify the study of internal models in a BMI context, we first asked whether subjects show evidence of internal prediction during BMI control. Next, we asked whether interpreting M1 activity through extracted internal models could explain movement errors that are present throughout proficient BMI control and long-standing deficiencies in control of BMI movement speed. Finally, because a key feature of internal models is their ability to adapt (*Shadmehr et al., 2010*), we altered the BMI mapping and asked whether the internal model adapted in a manner consistent with the new BMI mapping.

An important distinction that we make relative to previous work is that we are not asking circuit-level questions about how and where in the brain these internal models operate. Rather, we seek a statistical representation of the subject's prior beliefs about the BMI mapping (i.e., an internal model) that can be used to explain behavioral errors. Although internal models might not reside in M1 (*Shadmehr, 1997*; *Pasalar et al., 2006*; *Miall et al., 2007*; *Mulliken et al., 2008*; *Lisberger, 2009*), their computations influence activity in M1. Thus, by examining the moment-by-moment relationship between M1 population activity and task objectives, it may be possible to extract a detailed representation of the subject's internal model.

## Results

We trained two rhesus monkeys to modulate neural activity to drive movements of a computer cursor to hit targets in a two-dimensional workspace (*Figure 1B*). The family of BMI mappings that we used is represented by:

$$\mathbf{x}_t = \mathbf{A}\mathbf{x}_{t-1} + \mathbf{B}\mathbf{u}_t + \mathbf{b} \tag{1}$$

where $\mathbf{x}_t$ is the cursor state (position and velocity), $\mathbf{u}_t$ comprises the recorded M1 activity, and $\mathbf{A}$, $\mathbf{B}$, and $\mathbf{b}$ are the parameters of the BMI mapping. All experiments began with a closed-loop calibration of an *intuitive BMI mapping*, which was designed to provide proficient control on par with the majority of studies in the field (*Serruya et al., 2002*; *Velliste et al., 2008*; *Ganguly and Carmena, 2009*; *Suminski et al., 2010*; *Hauschild et al., 2012*; *Ifft et al., 2013*; *Sadtler et al., 2014*). Subjects indeed demonstrated proficient and stable control of the BMI, with success rates of nearly 100%, and movement times on average faster than one second (*Figure 1—figure supplement 1*).

The BMI provides an ideal paradigm for studying internal models because it simplifies several key complexities of native limb control. First, native limb control involves effectors with non-linear dynamics, and the causal relationship between the recorded neural activity and limb movements is not completely understood. In contrast, the causal relationship between recorded neural activity and BMI cursor movements is completely specified by the experimenter (through $\mathbf{A}$, $\mathbf{B}$ and $\mathbf{b}$ in *Equation 1*), and can be chosen to be linear (as in *Equation 1*). Second, native limb control involves multiple modalities of sensory feedback (e.g., proprioception and vision), which makes it difficult for the experimenter to know how the subject combines sources of sensory information. In the BMI, task-relevant sensory feedback is limited to a single modality (vision), which is completely specified by the experimenter ($\mathbf{x}_t$ in *Equation 1*). Finally, the neural activity that directly drives the BMI is completely specified by the recorded population activity ($\mathbf{u}_t$ in *Equation 1*), whereas typically only a subset of neurons driving limb movements is recorded. We can thereby reinterpret the full set of BMI control signals using an internal model in a more concrete manner than is currently possible with limb movements.

### Subjects compensate for sensory feedback delays while controlling a BMI

Because internal models have not previously been studied in a BMI context, we sought evidence of internal prediction. A hallmark of internal prediction is compensation for sensory feedback delays (*Miall et al., 2007*; *Shadmehr et al., 2010*; *Farshchiansadegh et al., 2015*). To assess the visuomotor latency experienced by a subject in our BMI system, we measured the elapsed time between target onset and the appearance of target-related activity in the recorded neural population (*Figure 2A*). The delays we measured (100 ms, monkey A; 133 ms, monkey C) are consistent with

visuomotor latencies reported in arm reaching studies of single-neurons in primary motor cortex (*Schwartz et al., 1988*). Next, we asked whether subjects produced motor commands consistent with the current cursor position, which was not known to the subject due to visual feedback delay, or whether motor commands were more consistent with a previous, perceived position (*Figure 2B,C* and *Figure 2—figure supplement 1*). If subjects did not compensate for visual feedback delays and aimed from the most recently available visual feedback of cursor position, we would expect errors to be smallest at lags of 100 ms and 133 ms relative to the current cursor position for monkeys A and C, respectively (dashed red lines in *Figure 2C*). Rather, we found that these error curves had minima at lags close to 0 ms (dashed black lines in *Figure 2C*), indicating that motor commands through the BMI mapping pointed closer to the targets when originating from the current cursor position than from any previous position. This finding suggests that subjects use an internal model to internally predict the current cursor position.

Because we have not yet explicitly identified the subject's *internal* model, motor commands were defined in this analysis using the BMI mapping, which is *external* to the subject. If the internal model bears similarities to the BMI mapping, it is reasonable to use the BMI mapping as a proxy for the internal model to assess feedback delay compensation. With evidence that subjects engage an internal model during BMI control, we next asked whether we could explicitly identify an internal model from the recorded neural activity.

## Internal model mismatch explains the majority of subjects' control errors

The BMI mapping, which determines the cursor movements displayed to the subject, provides one relevant, low-dimensional projection of the high-dimensional neural activity. With evidence that subjects use an internal model during closed-loop BMI control, we asked whether mismatch between an internal model and the actual BMI mapping could explain the subject's moment-by-moment aiming errors. This requires identifying the subject's internal model, which could reveal a different projection of the high-dimensional neural activity, representing the subject's internal beliefs about the cursor state. Because of the closed-loop nature of the BMI paradigm, the subject continually updates motor control decisions as new visual feedback of the cursor becomes available. To resolve these effects, the internal model needs to operate on a timescale of tens of milliseconds (in this case, a single timestep of the BMI system) on individual experimental trials. The extraction of such a rich internal model has been difficult prior to this study due to the lack of an appropriate statistical framework.

To overcome this limitation, we developed an internal model estimation (IME) framework, which extracts, from recorded population activity, a fully parameterized internal model along with a moment-by-moment account of the internal prediction process (*Figure 3A*). In the IME framework, the subject internally predicts the cursor state according to:

$$\tilde{\mathbf{x}}_{\mathbf{t}} = \tilde{\mathbf{A}}\tilde{\mathbf{x}}_{\mathbf{t}-1} + \tilde{\mathbf{B}}u_t + \tilde{\mathbf{b}} \qquad (2)$$

where $\tilde{\mathbf{x}}_t$ is the subject's internal prediction about the cursor state (position and velocity), $\mathbf{u}_t$ is a vector of recorded neural activity, and $\tilde{\mathbf{A}}$, $\tilde{\mathbf{B}}$, and $\tilde{\mathbf{b}}$ are the parameters of the subject's internal model. This form of the internal model was chosen to be analogous to the BMI mapping from *Equation 1* so that the actual BMI mapping lies within the family of internal models that we consider. Additionally, this formulation aligns with recent studies of skeletomotor (*Shadmehr and Krakauer, 2008*) and oculomotor (*Frens, 2009*) control, and a vast literature of control theory (*Anderson and Moore, 1990*).

The primary concept of the IME framework is that, at each timestep, the subject internally predicts the current cursor state by recursively applying *Equation 2* (starting from the most recently available sensory feedback) and generates neural activity consistent with aiming straight to the target relative to this internal prediction (see the 'Framework for internal model estimation (IME)' subsection in 'Materials and methods' and *Figure 3—figure supplement 1*). At each timestep, IME extracts the entire time-evolution of the subject's internal state prediction using *Equation 2* as an internal forward model. This evolution can be visualized in the form of a whisker (*Figure 3B*) that begins at the cursor position of the most recently available feedback and unfolds according to the extracted internal model. At each new timestep, the subject forms a new internal prediction that incorporates newly received visual feedback. If the internal model exactly matches the BMI mapping,

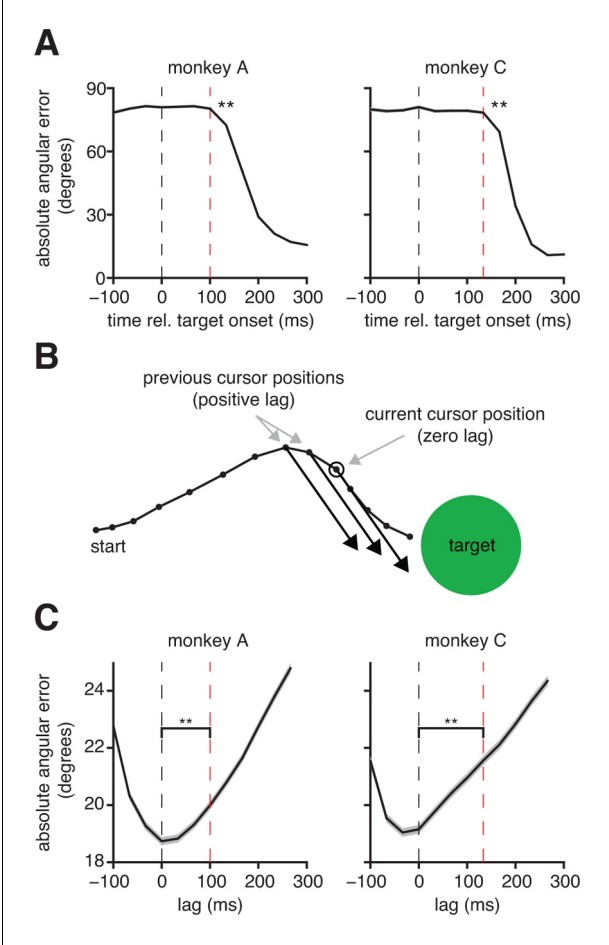

**Figure 2.** Subjects compensate for sensory feedback delays while controlling a BMI. (**A**) The visuomotor latency experienced by a subject in our BMI system was assessed by measuring the elapsed time between target onset and the first significant ($p < 0.05$) decrease in angular error. If that first decrease was detected $\tau + 1$ timesteps following target onset, we concluded that the visuomotor latency was at least $\tau$ timesteps (red dashed lines). For both subjects, the first significant difference was highly significant (**$p < 10^{-5}$, two-sided Wilcoxon test with Holm-Bonferroni correction for multiple comparisons; $n = 5908$ trials; monkey C: $n = 4578$ trials). (**B**) Conceptual illustration of a single motor command (black arrows) shifted to originate from positions lagged relative to the current cursor position (open circle). In this example, the command points farther from the target as it is shifted to originate from earlier cursor positions. (**C**) Motor commands pointed closer to the target when originating from the current cursor position (zero lag) than from outdated (positive lag) cursor positions that could be known from visual feedback alone (**$p < 10^{-5}$, two-sided Wilcoxon test; monkey A: $n = 33,660$ timesteps across 4489 trials; monkey C: $n = 31,214$ timesteps across 3639 trials). Red lines indicate subjects' inherent visual feedback delays from panel A. Shaded regions in panels A and C (barely visible) indicate $\pm$ SEM.

The following figure supplement is available for figure 2:

**Figure supplement 1.** Error metrics for assessing estimates of movement intent.

the subject's internal predictions would exactly match the cursor trajectory. A visualization of an example internal model and BMI mapping is given in *Figure 3—figure supplement 2*.

The central hypothesis in this study is that movement errors arise from a mismatch between the subject's internal model of the BMI and the actual BMI mapping. The alternative to this hypothesis is that the subject's internal model is well-matched to the BMI mapping, and movement errors result

from other factors, such as "noise" in the sensorimotor system, subjects' inability to produce certain patterns of neural activity, or subjects disengaging from the task. Our key finding is that recorded neural commands were markedly more consistent with the task goals when interpreted through subjects' internal models than when viewed through the BMI mapping (*Figure 3C*). Subjects' internal models deviated from the actual BMI mappings such that control errors evaluated through extracted internal models were substantially smaller than actual cursor errors: extracted internal models explained roughly 65% of cursor movement errors (70%, monkey A; 59%, monkey C). Although this finding does not preclude other factors (e.g., spiking noise or subject disengagement) from contributing toward movement errors, it does suggest their contribution is substantially smaller than previously thought, due to the large effect of internal model mismatch.

We found that the majority of the explanatory power of extracted internal models was in their ability to identify structure in the high-dimensional neural activity (*Figure 3—figure supplement 3*). This structure was captured in the internal model by the mapping from high-dimensional neural activity to low-dimensional kinematics ($\tilde{\mathbf{B}}$ in *Equation 2*), which need not match the BMI mapping ($\mathbf{B}$ in *Equation 1*). Consistent with this finding, internal models fit to low-dimensional behavior rather than high-dimensional neural activity were not able to explain cursor errors (*Figure 3—figure supplement 4*).

That a majority of cursor errors can be explained by mismatch of the internal model is not to say that control through the BMI mapping was poor–in fact control was proficient and stable (*Figure 1B* and *Figure 1—figure supplement 1*). Rather, extracted internal models predicted movements that consistently pointed straight to the target, regardless of whether the actual cursor movements did (*Figure 4A*) or did not (*Figure 4B* and *Figure 4—figure supplement 1*) point straight to the target. On most trials, BMI cursor trajectories proceeded roughly straight to the target (*Figure 4A*). On these trials, internal model predictions aligned with actual cursor movements, resulting in small errors through both the BMI mapping and the extracted internal model. In a smaller subset of trials, actual cursor movements were more circuitous and thus had relatively large errors. Previously, the reason behind these seemingly incorrect movements was unknown, and one possibility was that the subject simply disengaged from the task. When interpreted through the extracted internal model, however, neural activity during these circuitous trials appears correct, suggesting that the subject was engaged but was acting under an internal model that was mismatched to the BMI mapping (*Figure 4B* and *Figure 4—figure supplement 1*). In other words, when armed with knowledge of the subject's internal model, outwardly irrational behavior (i.e., circuitous cursor movements) appears remarkably rational. Across all trials, the majority of neural activity patterns had low or zero error as evaluated through extracted internal models, regardless of whether errors of the actual cursor movements (i.e., through the BMI mapping) were large or small (*Figure 4C* and *Figure 4—figure supplement 2*).

When cursor trajectories were circuitous, it was not uncommon for some internal model predictions (whiskers) to match the actual cursor movement while others did not, even within the same trial (*Figure 4B*). Given a single internal model, how can some patterns of neural activity result in whiskers aligned to the cursor trajectory, while others patterns produce whiskers that deviate from the cursor trajectory? This is possible due to mathematical operation of mapping from high-dimensional neural activity patterns to low-dimensional cursor states. *Figure 4D* provides a conceptual illustration of a simplified BMI mapping:

$$\mathbf{v}_t = \mathbf{B}\mathbf{u}_t \tag{3}$$

and a simplified internal model:

$$\tilde{\mathbf{v}}_t = \tilde{\mathbf{B}}\mathbf{u}_t \tag{4}$$

each of which relies only on a mapping ($\mathbf{B}$ or $\tilde{\mathbf{B}}$) from neural activity ($\mathbf{u}_t$) to cursor velocity ($\mathbf{v}_t$ or $\tilde{\mathbf{v}}_t$). We focus on $\mathbf{B}$ and $\tilde{\mathbf{B}}$ here (without considering $\mathbf{A}$, $\mathbf{b}$, $\tilde{\mathbf{A}}$, and $\tilde{\mathbf{b}}$ from *Equations 1* and *2*) because of the aforementioned finding that the majority of the internal model mismatch effect is captured by differences between $\mathbf{B}$ and $\tilde{\mathbf{B}}$ (*Figure 3—figure supplement 3*). Given a mismatched BMI mapping (black lines) and internal model (red lines), many neural activity patterns will produce different velocities through the BMI mapping versus the internal model. However, a subset of activity patterns (gray line) will produce identical velocities through both the BMI mapping and the internal model. These

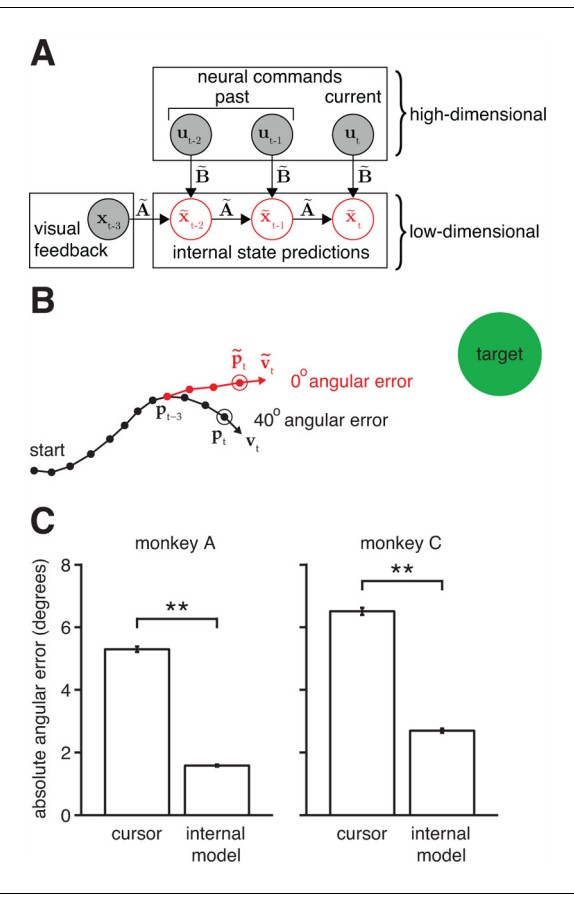

**Figure 3.** Mismatch between the internal model and the BMI mapping explains the majority of the subjects' cursor movement errors. (**A**) At each timestep, the subject's internal state predictions ($\tilde{\mathbf{x}}_{t-2}$, $\tilde{\mathbf{x}}_{t-1}$, $\tilde{\mathbf{x}}_t$) are formed by integrating the visual feedback ($\mathbf{x}_{t-3}$) with the recently issued neural commands ($\mathbf{u}_{t-2}$, $\mathbf{u}_{t-1}$, $\mathbf{u}_t$) using the internal model ($\tilde{\mathbf{A}}, \tilde{\mathbf{B}}, \tilde{\mathbf{b}}$). We defined cursor states and internal state predictions to include components for position and velocity (i.e., $\mathbf{x}_t = [\mathbf{p}_t; \mathbf{v}_t]$, $\tilde{\mathbf{x}}_t = [\tilde{\mathbf{p}}_t; \tilde{\mathbf{v}}_t]$). (**B**) Cursor trajectory (black line) from a BMI trial that was not used in model fitting. Red *whisker* shows the subject's internal predictions of cursor state as extracted by IME. The critical comparison is between the actual cursor velocity ($\mathbf{v}_t$; black arrow) and the subject's internal prediction of cursor velocity ($\tilde{\mathbf{v}}_t$; red arrow). (**C**) Cross-validated angular aiming errors based on IME-extracted internal models are significantly smaller than cursor errors from the BMI mapping (**$p < 10^{-5}$, two-sided Wilcoxon test; monkey A: $n$ = 5908 trials; monkey C: $n$ = 4577 trials). Errors in panel B are from a single timestep within a single trial. Errors in panel C are averaged across timesteps and trials. Errors in panels B and C incorporate temporal smoothing through the definition of the BMI mapping and the internal model, and are thus not directly comparable to the errors shown in *Figure 2C*, which are based on single-timestep velocity commands needed for additional temporal resolution. Error bars (barely visible) indicate ± SEM.

The following figure supplements are available for figure 3:

**Figure supplement 1.** Full probabilistic graphical model for the internal model estimation (IME) framework.

**Figure supplement 2.** A unit-by-unit comparison of the subject's internal model and the BMI mapping.

**Figure supplement 3.** The explanatory power of IME comes primarily from structure in the high-dimensional neural activity.

**Figure supplement 4.** IME does not explain cursor errors when fit to low-dimensional behavior.

**Figure supplement 5.** Subjects could readily produce the entire range of movement directions through the BMI mapping.

*Figure 3 continued*

**Figure supplement 6.** Internal model mismatch is not an artifact of correlated spiking variability.

**Figure supplement 7.** IME does not explain cursor errors when fit to neural commands that do not contain high-dimensional structure.

**Figure supplement 8.** A simplified alternative internal model is not consistent with the data.

patterns lie in the nullspace of $\mathbf{B} - \tilde{\mathbf{B}}$ (i.e., solutions to the equation $\mathbf{B}\mathbf{u}_t = \tilde{\mathbf{B}}\mathbf{u}_t$). In the example trials shown in *Figure 4A,B* and *Figure 4—figure supplement 1*, internal model predictions (red) that match the actual cursor movement (black) correspond to neural activity patterns along the gray line in *Figure 4D*. Predictions not matching the cursor movement correspond to neural activity patterns anywhere off the gray line in *Figure 4D*.

## Two alternative hypotheses do not explain the effect of internal model mismatch

The data presented thus far support our central hypothesis that internal model mismatch is a primary source of movement errors. Next we asked whether it might be possible to have arrived at this result under the alternate hypothesis that the internal model is well-matched to the BMI mapping. We address two specific cases of this alternative hypothesis and show that they do not explain the observed effect of internal model mismatch.

First, we explored the possibility that the subject might have a well-matched internal model, but has systematic difficulties producing the neural activity patterns required to drive the cursor in all directions in the 2D workspace using the BMI mapping. This could result in an estimated internal model that appears to be mismatched to the BMI mapping. Although M1 cannot readily produce all possible patterns of high-dimensional neural activity (*Sadtler et al., 2014*), we observed that subjects could readily produce the full range of movement directions through the BMI mapping (*Figure 3—figure supplement 5*). Gaps between producible movement directions were typically less than 1/4 of a degree, which is substantially smaller than the cursor errors shown in *Figure 3C*. This suggests that our main finding of internal model mismatch cannot be explained by subjects' inability to produce particular neural activity patterns.

Second, we explored the possibility that the subject intended to produce neural commands that were correct according to the BMI mapping, but that those intended commands were corrupted by "noise" that is oriented such that errors appear smaller through the extracted internal model than through the BMI mapping. Here we define noise as spiking variability not explained by the desired movement direction under the BMI mapping. If spiking variability is correlated across neurons, it is possible to identify a mapping that best attenuates that variability. To determine whether correlated spiking variability could explain the effect of internal model mismatch, we simulated neural activity according to this alternative hypothesis in a manner that preserved the statistics of the real data (*Figure 3—figure supplement 6*). If this simulation produced results that match our findings from the real data, it would indicate that our main finding can be explained by the alternate hypothesis. However, this was not the case. Simulated neural activity was more consistent with the BMI mapping than the extracted internal model, which contrasts with our finding from the recorded neural activity.

## Statistical controls for validating observed effects

To further validate the main results presented above, we implemented four statistical controls. First, we ensured that our findings were not simply artifacts of overfitting the data. Second, we removed the high-dimensional structure from the neural activity while preserving the cursor movements, and show that resulting extracted internal models no longer provided explanatory power. Third, we ensured that internal model predictions do not trivially point toward the targets. Finally, we explored a variety of forms for the internal model and found that a simplified form does not account for the data. Here we describe each of these four statistical controls in additional detail.

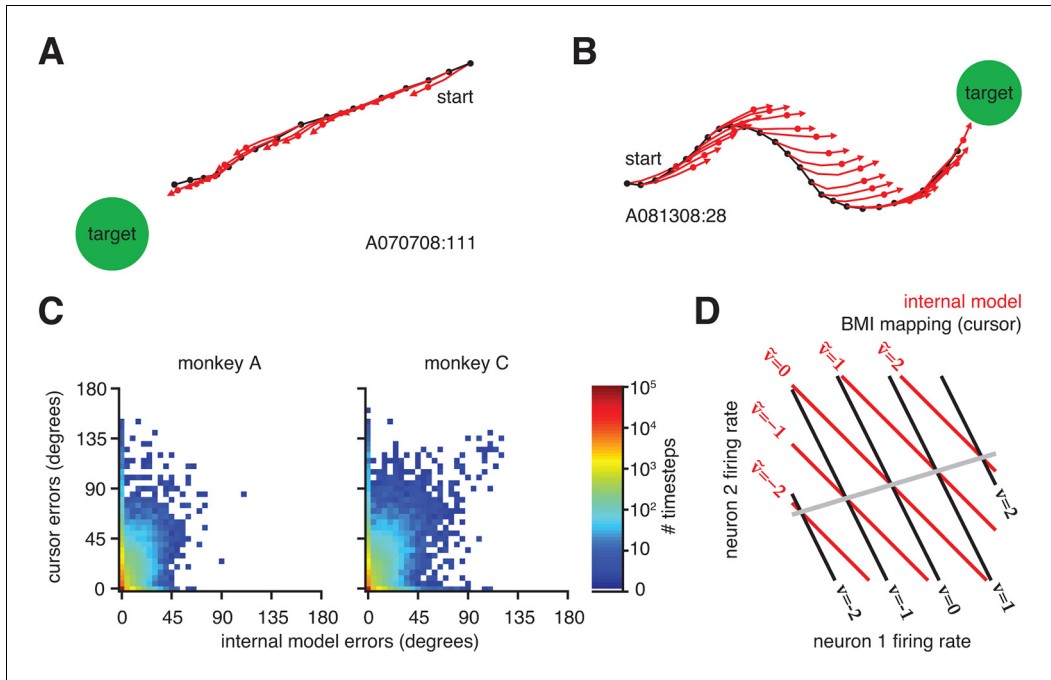

**Figure 4.** Neural activity appears correct through the internal model, regardless of how the actual cursor moved. (**A**) Typical trial in which the cursor followed a direct path (black) to the target. Internal model predictions (red whiskers) also point straight to the target. (**B**) Trial with a circuitous cursor trajectory. Internal model predictions point straight to the target throughout the trial, regardless of the cursor movement direction (same color conventions as in panel A). (**C**) Timestep-by-timestep distribution of BMI cursor and internal model errors. Neural activity at most timesteps produced near-zero error through the internal model, despite having a range of errors through the BMI mapping. (**D**) Hypothetical internal model (red) and BMI mapping (black) relating 2D neural activity to a 1D velocity output. This is a simplified visualization of *Equations 1* and *2*, involving only the $\mathbf{B}$ and $\tilde{\mathbf{B}}$ parameters, respectively. Each contour represents activity patterns producing the same velocity through the internal model ($\tilde{\mathbf{v}}$, red) or BMI mapping ($\mathbf{v}$, black). Because of internal model mismatch, many patterns result in different outputs through the internal model and the BMI. However, some patterns result in the same output through both the internal model and the BMI (gray line). Here we illustrate using a 2D neural space and 1D velocity space. In experiments with $q$-dimensional neural activity and 2D velocity, activity patterns producing identical velocities through both the internal model and the cursor span a $(q-4)$-dimensional space.

The following figure supplements are available for figure 4:

**Figure supplement 1.** IME whiskers consistently point to the target regardless of cursor movement direction.

**Figure supplement 2.** Errors from trials in *Figure 4—figure supplement 1* highlighted on the distribution of errors across trials.

One possible concern when interpreting the findings presented above is that internal models might be simply overfitting the data. To rule out this possibility, all findings presented throughout this paper are cross-validated (see the 'Computing cross-validated internal model predictions' subsection in 'Materials and methods'). Internal models were fit using a subset of trials as training data. Then, trials that were held out during fitting were used to evaluate each extracted internal model. If the extracted internal models had overfit the training data, we would expect those internal models to generalize poorly to the held-out data. However, this was not the case. Internal models explained the majority of cursor errors in the held-out data (*Figure 3C*), demonstrating that extracted internal models captured real, task-relevant structure in the recorded neural activity.

In addition to properly cross-validating our results, we performed a control analysis to show that extracted internal models identified reliable, task-appropriate structure in the high-dimensional neural activity. Here we extracted internal models using neural activity that had been shuffled across

timesteps in a manner that preserved the cursor movements through the BMI mapping (*Figure 3—figure supplement 7*). If our results could be explained by internal models that simply overfit noise in the data, we would expect internal models fit to these shuffled data data sets to again explain a majority of cursor errors. However, internal models extracted from these shuffled data sets could no longer explain cursor errors, indicating that IME does not identify effects when they do not exist in the data. This result is consistent with our findings that the majority of the explanatory power of extracted internal models relies on structure in the high-dimensional neural activity (*Figure 3—figure supplement 3*), and that cursor errors cannot be explained by internal models when high-dimensional neural activity is replaced by low-dimensional behavioral measurements during model fitting (*Figure 3—figure supplement 4*).

If an internal model prediction points toward the target, it is not trivially due to our inclusion of straight-to-target aiming during model fitting (see the 'Computing cross-validated internal model predictions' subsection in 'Materials and methods'). Although target positions were used during model fitting, they were never used when computing internal model predictions from the data (e.g., when constructing the whiskers in *Figure 3B*, *Figure 4A,B*, and *Figure 4—figure supplement 1*). Each whisker was constructed in a held-out trial using only visual feedback (consisting of a single timestep of cursor position and velocity), the recorded neural activity up through the current timestep, and the internal model extracted from the training data. Because of our aforementioned cross-validation procedures, when the neural command $\mathbf{u}_t$ is used to compute the movement error at timestep $t$, that neural command had not been seen previously (i.e., it was not used when fitting the internal model, when estimating the subject's internal cursor state prediction, when calibrating the BMI mapping, nor when determining the current position of the actual BMI cursor). A whisker that points straight to the target in the held-out data thus reveals that, when interpreted through the subject's internal model, the recorded neural activity would have driven the cursor straight to the target.

Finally, we explored a variety of approaches to modeling the subject's internal tracking process and found that models demonstrated similarly high degrees of explanatory power as long as they could capture high-dimensional structure in the neural activity. However, a simplified internal model that does not account for any form of internal forward prediction was not consistent with our data (*Figure 3—figure supplement 8*).

## Internal model mismatch explains limitations in speed dynamic range

A major limitation in BMI performance is the ability to control cursor speed (*Gilja et al., 2012*; *Golub et al., 2014*). *Gilja et al. (2012)* and *Golub et al. (2014)* have proposed solutions to improve control of BMI speed (in particular, with respect to stopping the BMI cursor at targets). However, it is still an open question as to why BMI speed control is deficient in the first place. In addition to explaining the subjects' aiming errors, we asked whether mismatch between the internal model and BMI mapping could also explain subjects' difficulty in controlling cursor speed. Using the extracted internal model, we could compare the subject's intended speed (from the internal model) to the speed of the actual BMI cursor at each timestep. We found that low intended speeds were systematically overestimated, and high intended speeds were systematically underestimated by the BMI mapping (*Figure 5A*). Furthermore, we discovered that the subjects intended to hold the cursor steadier during the initial hold period and move the cursor faster during the movement than what occurred during experiments (*Figure 5B*). Note that we make no assumptions about movement speed when extracting the internal model (see the 'Framework for internal model estimation (IME)' subsection in 'Materials and methods').

To gain insight into this speed mismatch, we can use extracted internal models to examine the discrepancies between intended and actual speeds at the level of individual units and on the timescale of a single 33-ms timestep (*Figure 5—figure supplement 1*). These systematic differences between intended and actual cursor speeds indicate that internal model mismatch limits realizable dynamic range of BMI movement speeds. These findings suggest that the longstanding deficiencies in BMI speed control may be a consequence of internal model mismatch.

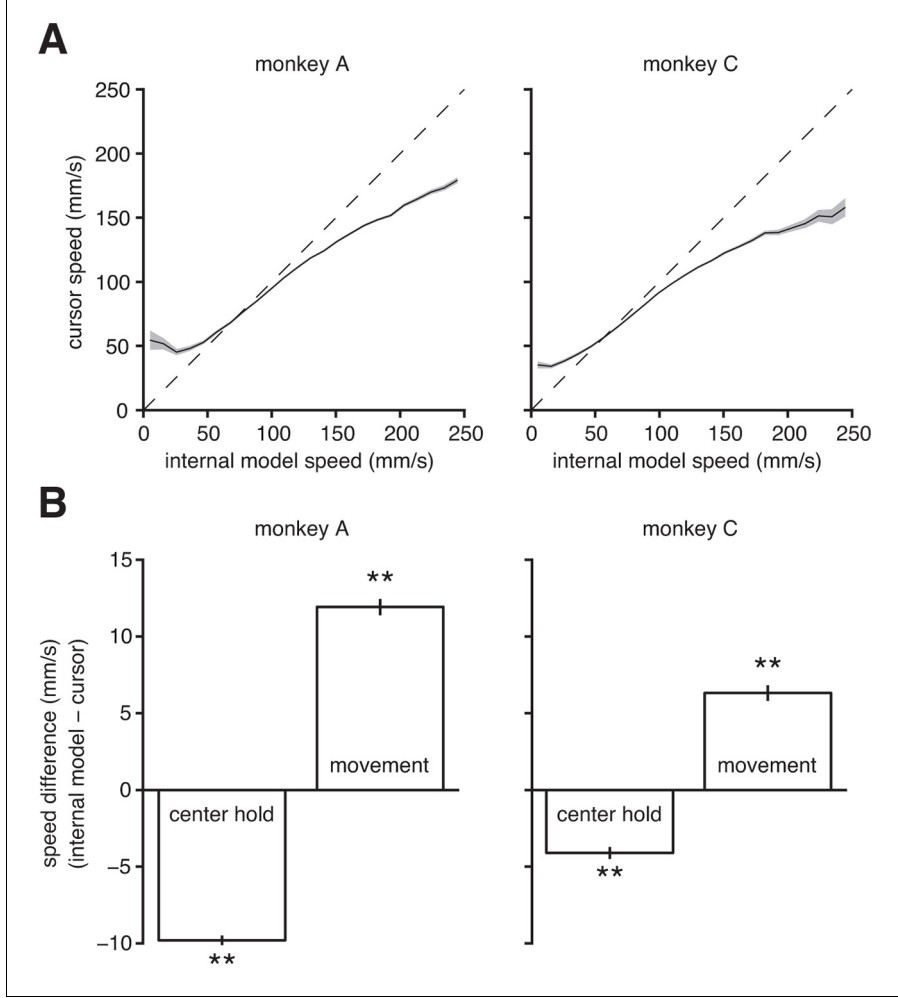

**Figure 5.** Internal model mismatch limits the dynamic range of BMI cursor speeds. (**A**) BMI cursor speeds across the range of intended (i.e., internal model) speeds. At low intended speeds, BMI speeds were higher than intended, whereas for mid-to-high intended speeds, BMI speeds were lower than intended. Shaded regions indicate ± SEM. (**B**) During the hold period prior to target onset, intended speeds were significantly lower than those produced through the BMI mapping. During movement, intended speeds were significantly higher than those produced through the BMI. Error bars indicate ± SEM (**$p < 10^{-5}$, two-sided Wilcoxon test; monkey A: $n = \{5006, 5908\}$ trials; monkey C: $n = \{3008, 4578\}$ trials). In panels A and B internal models were used to predict intended speed on trials not used during model fitting.

The following figure supplement is available for figure 5:

**Figure supplement 1.** A unit-by-unit example of internal model mismatch limiting cursor speed dynamic range.

## Perturbations drive internal model adaptation

A key feature of an internal model is its ability to adapt. Arm reaching studies have demonstrated behavioral evidence of internal model adaptation (*Shadmehr and Mussa-Ivaldi, 1994*; *Thoroughman and Shadmehr, 2000*; *Joiner and Smith, 2008*; *Taylor et al., 2014*). Behavioral learning has also been demonstrated in the context of BMIs (*Taylor, 2002*; *Carmena et al., 2003*; *Jarosiewicz et al., 2008*; *Ganguly and Carmena, 2009*; *Chase et al., 2012*; *Sadtler et al., 2014*). While these BMI studies suggest that subjects adapt their internal models to better match the BMI mapping, a direct assessment has been difficult without access to those internal models. With the ability to extract a subject's internal model, here we asked whether extracted internal models adapt in accordance with perturbations to the BMI mapping (*Figure 6*). In one monkey, an initial block of

trials under an intuitive BMI mapping was followed by a block of trials under a perturbed BMI mapping. All data analyzed prior to this section were recorded during intuitive trials. The intuitive and perturbed mappings were of the form of *Equation 1*, but each used different values in the matrix **B**. The perturbed BMI mapping effectively rotated the pushing directions of a subset of the recorded units, such that the global effect resembled a visuomotor rotation (see the 'Behavioral task' subsection in 'Materials and methods'). Previous studies have shown that perturbations of this type can be learned by monkeys (*Wise et al., 1998*; *Paz et al., 2005*; *Chase et al., 2012*).

For each experiment, we interpreted recorded population activity through the intuitive and perturbed BMI mappings, as well as through two views of the subject's internal model: a time-varying internal model extracted from a moving window of 48 trials, and a late intuitive internal model extracted from the last 48 intuitive trials. We could then quantify changes in the subject's internal

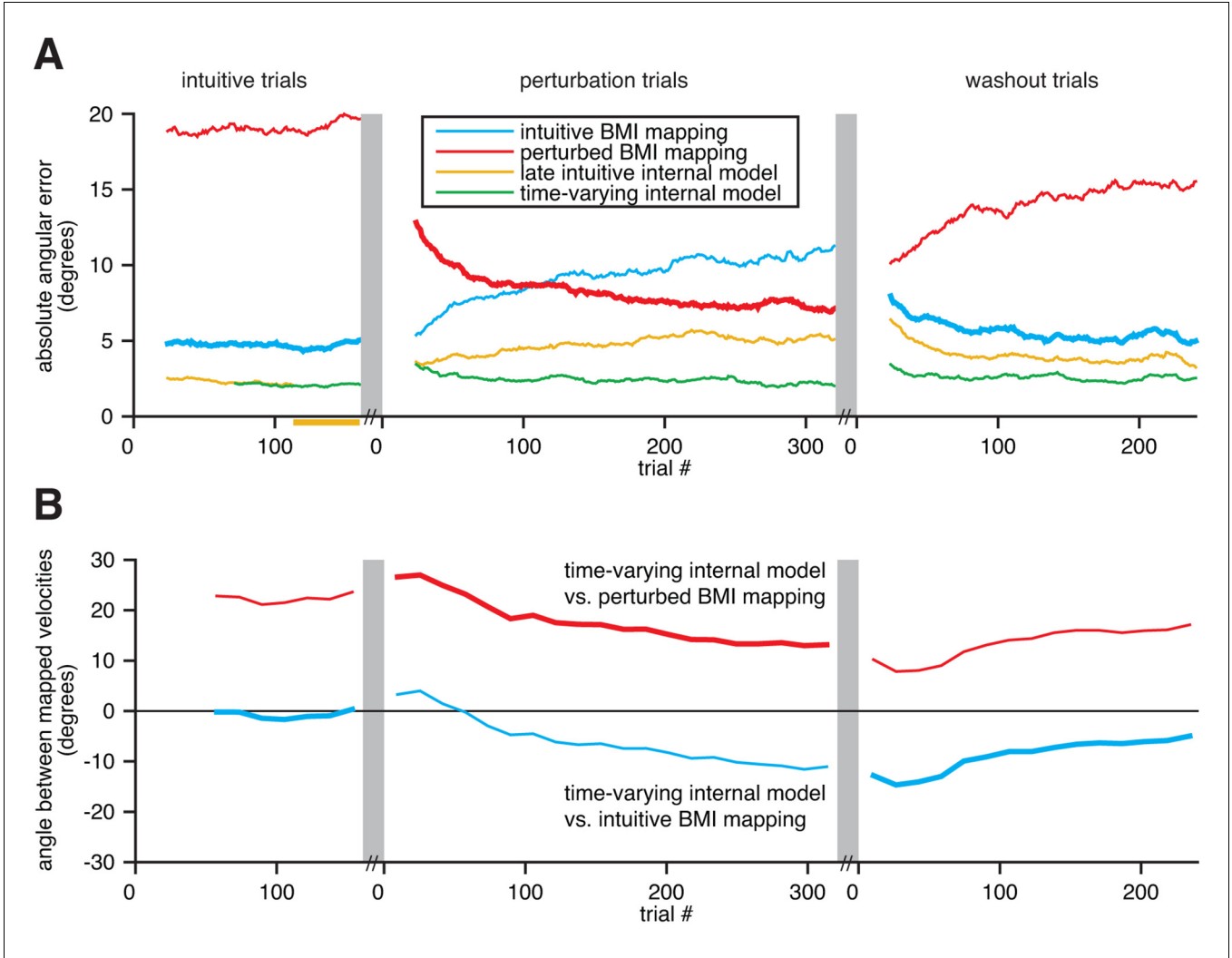

**Figure 6.** Extracted internal models capture adaptation to perturbations. (**A**) Cross-validated angular errors computed by interpreting monkey A neural activity through BMI mappings and internal models. The intuitive BMI mapping (blue) defined cursor behavior during the intuitive and washout trials. The perturbed BMI mapping (red) defined cursor behavior during the perturbation trials. The late intuitive internal model (yellow) was extracted from the last 48 intuitive trials (yellow bar). A time-varying internal model (green) was extracted from a moving window of the 48 preceding trials. Values were smoothed using a causal 24-trial boxcar filter and averaged across 36 experiments. (**B**) Differences between monkey A's time-varying internal model and the BMI mappings, assessed through the high-dimensional neural activity. For each round of 16 trials, neural activity from those trials was mapped to velocity through the time-varying internal model, the intuitive BMI mapping, and the perturbed BMI mapping. Signed angles were taken between velocities computed through the time-varying internal model and the intuitive BMI mapping (blue) and between velocities computed through the time varying internal model and the perturbed BMI mapping (red). Values were averaged across 36 experiments.

model and assess which BMI mapping or internal model was most consistent with the neural activity, relative to task goals (*Figure 6A*). To avoid circularity, trials used to evaluate the BMI mappings and internal models were not used when fitting the internal models nor when calibrating the BMI mappings.

Errors through the intuitive BMI mapping describe the actual cursor performance during the intuitive and washout trials (thick blue traces; analogous to cursor errors in *Figure 3C*), and how that mapping would have performed had it been in effect during the perturbation trials (thin blue trace). Similarly, errors through the perturbed BMI mapping describe the actual cursor performance during the perturbation trials (thick red trace), and how that mapping would have performed had it been in effect during the intuitive and washout trials (thin red traces). Behavioral learning was evident in that errors through the perturbed BMI mapping were large in early perturbation trials and decreased continuously throughout the perturbation trials. A detailed characterization of this behavioral learning can be found in (*Chase et al., 2012*).

Our key finding in this analysis is that extracted internal models adapted in a manner consistent with the BMI perturbations (*Figure 6B*). During the perturbation trials, the time-varying internal model adapted to better match the perturbed BMI mapping (red trace trends toward zero). Similarly, during the washout trials, the time-varying internal model adapted to better match the intuitive BMI mapping (blue trace trends toward zero). Had the subject's internal model not adapted, or if the adaptation was not reflected in the extracted internal model, we would expect the traces in *Figure 6B* to be flat. Rather than being static entities, the extracted internal models were dynamic with timescales independent of experimenter-induced changes to the BMI mapping.

Consistent with our central hypothesis, internal model mismatch was present throughout the intuitive, perturbation, and washout trials. During intuitive trials, errors through the time-varying internal model were substantially lower than errors through the intuitive BMI mapping (green trace lower than blue trace in *Figure 6A*), which is consistent with our main findings in *Figure 3C*. Because the subject's internal model adapts, errors through the time-varying internal model remained substantially smaller than errors through the BMI mappings across the perturbation and washout trials as well (green trace remains low across *Figure 6A*). Although behavioral errors decreased over the course of the perturbation and washout trials, internal model mismatch was still present following adaptation (red and blue traces are nonzero during late perturbation and washout trials, respectively, in *Figure 6B*).

It could have been that this internal model mismatch was only substantial during early intuitive trials before the subject had accrued enough experience to form a stable internal model. This was not the case. The subject's internal model was stable throughout the intuitive session, as evidenced by the nearly constant angular differences between velocities mapped through the time-varying internal model and the BMI mapping (red and blue traces are roughly flat in *Figure 6B*) and the nearly identical errors through the time-varying internal model and the (static) late intuitive internal model (green and yellow traces overlap in *Figure 6A* wherever cross-validated errors can be computed). Consistent with a stable internal model, behavioral performance was stable throughout the intuitive trials (blue trace is flat in *Figure 6A*), and internal model mismatch explained a steady fraction of behavioral errors (green trace is also flat, and substantially lower than blue trace). During the perturbation session, the subject's internal model diverges from this stabilized state (yellow trace diverges from green trace in *Figure 6A* and both traces are non-constant in *Figure 6B*).

## Discussion

In this work, we asked what gives rise to behavioral errors during feedback motor control. In a BMI paradigm, we hypothesized that a mismatch between the subject's internal model and the actual BMI mapping could explain errors in BMI cursor movement. To assess this, we first found evidence that subjects compensate for sensory feedback delays. Then, we reinterpreted the neural population activity recorded during closed-loop BMI control in terms of a rich internal model that operates on a timescale of tens of milliseconds. We found (i) that subjects' neural activity was often correct according to the internal model even when cursor movements were circuitous (thereby explaining 65% of cursor errors), and (ii) that subjects intended to hold the cursor steadier during initial hold periods and to drive the cursor faster during movements, relative to observed behavior. Furthermore, when

the BMI mapping was perturbed, the subject's internal model changed in a manner consistent with the new BMI mapping.

## Internal models influence activity in M1

In this study, we considered neural population activity recorded in M1. It is reasonable to ask how it is possible to deduce anything about internal models from M1 activity if we are not recording i) signals from the neural circuits that implement internal models (e.g., cerebellum), nor ii) the internal copy signals that enable internal model computations. The rationale is the following. First, the neural activity recorded in M1 is likely to be downstream of the internal model computations, whether they be in cerebellum (*Shadmehr, 1997*; *Pasalar et al., 2006*; *Miall et al., 2007*; *Lisberger, 2009*; *Huang et al., 2013*), posterior parietal cortex (*Shadmehr, 1997*; *Mulliken et al., 2008*), dorsal premotor cortex (*Shadmehr, 1997*), or elsewhere. Thus, the internal model is likely to influence the neural activity produced in M1. By relating the neural activity recorded in M1 to the behavioral task on a moment-by-moment basis, we should be able to infer properties of the upstream internal model. Second, previous studies indicate that internal copy signals (e.g., efference copy, corollary discharge) carry information about movement intent, in particular a copy of the movement intent that M1 sends to the motor effector (*Crapse and Sommer, 2008*; *Huang et al., 2013*; *Schneider et al., 2014*; *Azim et al., 2014*). Although we are not directly recording the internal copy signal, the information in the internal copy relevant to movement intent is likely also present in the recorded M1 activity, and this is what we leveraged. In short, we make no claims about the neural circuitry implementing internal models, but rather we infer statistical properties of the internal models from their downstream consequences in M1. Using this rationale, we extracted internal models from M1 population activity.

We chose to capture the subject's internal model using a forward model framework (*Equation 2* and *Figure 3A*) because it is both highly interpretable and consistent with a large body of behavioral and computational studies (*Shadmehr and Krakauer, 2008*; *Frens, 2009*). Our results do not preclude the use of other types of internal models, such as an inverse model (*Shadmehr and Mussa-Ivaldi, 1994*; *Kawato, 1999*), whose acquisition and function is believed to be tightly coupled to that of the forward model (*Wolpert and Kawato, 1998*).

## Avoiding circularity when extracting an internal model

We presented four important lines of evidence that indicate that the extracted internal models are meaningful, and not a result of logical circularity during model fitting or overfitting to noise in the data. First, extracted internal models explain a majority of behavioral errors on trials not seen during model fitting (*Figure 3C*). Here, extracted internal models identified structure in the high-dimensional neural activity that indicated straight-to-target movement intent, even when the cursor behavior was circuitous. Internal model predictions on held-out trials could not trivially point toward the targets because that held-out neural activity had not been used during model fitting, and because target positions were never used when constructing internal model predictions from held-out trials. Second, the finding that intended speed is better predicted by internal models than the BMI mapping (*Figure 5B*) lends an additional independent validation of those internal models, since no assumptions were made about intended movement speed when fitting internal models. Third, when we perturbed the BMI mapping, extracted internal models revealed adaptation consistent with the particular perturbations (*Figure 6*).

Finally, we performed a series of scientific and statistical control analyses. We showed that our data are not consistent with two versions of the alternative hypothesis, in which the subject's internal model is well-matched to the BMI mapping (*Figure 3—figure supplement 5* and *Figure 3—figure supplement 6*). Further, we asked whether extracted internal models could explain the observed behavioral errors without access to structure in the high-dimensional neural activity beyond that which defined cursor movements. We considered two different alterations to the data from which internal models were extracted: one in which we replaced the high-dimensional neural activity with low-dimensional cursor velocities (*Figure 3—figure supplement 4*) and another in which we shuffled the neural activity in a manner that preserved cursor velocities through the BMI mapping (*Figure 3—figure supplement 7*). In both cases, we found that the extracted internal models no longer offered

a consistent explanation for the observed behavioral errors, thereby demonstrating that the explanatory power of the extracted internal models does not arise from logical circularity or overfitting.

## Relationship between internal models and BMI mappings

An extracted internal model and a BMI mapping are closely related. They take a similar mathematical form (*Equations 1* and *2*) and both project high-dimensional population activity to a low-dimensional kinematic space. A key difference between internal models and BMI mappings is that internal models are dynamic entities whose properties can change during motor adaptation. In contrast, the BMI mappings are chosen by the experimenter or by a computer algorithm. Critically, in experiments in which we abruptly applied a perturbed BMI mapping, we found that extracted internal models dynamically adjusted in a manner appropriate for the task and at a timescale independent of changes to the BMI mapping (*Figure 6*). The ability to interpret neural activity through the subject's internal model, while the subject controls the cursor through some BMI mapping (e.g., *Figure 4A,B*, *Figure 6A* and *Figure 4—figure supplement 1*), offers a unique glimpse into the subject's movement intentions, sensory prediction errors, and motor adaptation.

Given the substantial fraction of behavioral errors that are explained by internal model mismatch during control under the intuitive BMI mapping, it is perhaps surprising that we did not find evidence of behavioral or internal model adaptation during those trials (*Figure 6*). A way to reconcile these findings is that, in contrast to the frequent movement errors experienced after the BMI mapping was perturbed, there was a relative paucity of errors during the intuitive trials. As a result, there may not have been sufficient pressure to improve upon a "good enough" internal model (*Loeb, 2012*). Had the subject been given more experience with the same BMI mapping (*Ganguly and Carmena, 2009*), the internal model may have converged to the BMI mapping. Nevertheless, our findings indicate that the subject's learning process may be a key limitation in BMI performance (*Sadtler et al., 2014*). It may be possible to overcome these limitations in the subject's neural adaptation process through complementary innovations in designing the BMI mapping (*Shenoy and Carmena, 2014*). For example, applying an extracted internal model as the BMI mapping might improve performance during closed-loop BMI control. Indeed, a recent study incorporating the concept of internal tracking has demonstrated substantial gains in closed-loop BMI performance (*Gilja et al., 2012*). Future studies will be required to determine whether further improvements in performance might be possible by using the IME framework toward designing the BMI mapping.

## Leveraging multi-dimensional structure in population activity

The insights gained in this study were made possible because we monitored the subject's high-dimensional neural activity. Because the BMI mapping and the subject's internal model are high-to-low dimensional mappings, neural activity that was consistently correct under the internal model sometimes resulted in aberrant behavior through the BMI mapping. We would not have been able to observe or explain this phenomenon by analyzing the BMI cursor movements in isolation (*Figure 3—figure supplement 4*). In particular, by replacing all instances of neural activity (i.e., the $\mathbf{u}_t$ in *Equation 2*) with actual cursor velocities (or analogously, with actual hand velocities from an arm reaching task), IME becomes limited to predicting the subject's velocity intent to be a scaled and rotated (in two-dimensions) version of the actual velocity. In contrast, access to the high-dimensional neural activity enabled the identification of the subject's intended movements without constraining them to have a consistent relationship with actual movements.

Prior beliefs, and their role in sensation and behavior, have been the focus of many studies, including those on visual perception (*Kersten et al., 2004*; *Komatsu, 2006*; *Berkes et al., 2011*), perceptual decision-making (*Ma and Jazayeri, 2014*), and sensorimotor learning (*Körding and Wolpert, 2004*; *Turnham et al., 2011*). Our work provides a means for extracting a rich representation of prior beliefs (i.e., the internal model) that can combine past sensory input with multi-dimensional neural processes to drive moment-by-moment motor control decisions. We found that outwardly aberrant behavior and behavioral limitations could be explained by taking into account the subject's prior beliefs. By recording simultaneously from multiple neurons and developing the appropriate statistical algorithms, it may be possible to extract similarly rich prior beliefs in other systems.

## Materials and methods

### Neural recordings

Two male rhesus macaques (Maccaca mulatta) were each implanted with a 96-channel microelectrode array (Blackrock Microsystems, Salt Lake City, UT) targeting proximal arm area of primary motor cortex. Signals were amplified, bandpass filtered (250 Hz - 8 kHz) and manually sorted (Plexon Sort Client, box sort) with a 96-channel Plexon MAP system (Plexon, Dallas, TX). Recorded neuronal units were either well-isolated single cells or multiple cells that could not be well separated but as a group were tuned to intended movement direction. In each session, we recorded $26.0 \pm 3.4$ (monkey A) and $39.2 \pm 3.9$ (monkey C) neuronal units (mean $\pm$ one standard deviation). Spike counts were taken in nonoverlapping 33-ms bins throughout the behavioral task (see 'Behavioral task'). All animal procedures were approved by the Institutional Animal Care and Use Committee of the University of Pittsburgh.

### Behavioral task

Subjects modulated neural activity to drive movements of a virtual cursor in a 2D brain-machine interface (BMI) task. The cursor (radius: 7–8 mm, monkey A; 6 mm monkey C) and targets (same radii as cursor) were displayed to the subject on a frontoparallel stereoscopic display (Dimension Technologies, Rochester, NY) with a refresh rate of 60 Hz. Display updates were subject to a latency of up to 2 refresh cycles (0–33.3 ms). Target positions were chosen pseudorandomly from a set of 16 evenly-spaced radial targets (center-to-target distance: 85 mm, monkey A; 72–73 mm, monkey C). Each trial began with the cursor at the workspace center, where the subject was required to hold the cursor to visibly overlap a central target (center hold requirement randomly selected for each trial: 50–350 ms, monkey A; 50–150 ms, monkey C). Following completion of the initial hold, a peripheral target appeared, instructing the subject to initiate a cursor movement. Target acquisition was recorded as the first timestep during which the cursor visibly overlapped the peripheral target. Following target acquisition, the subject was required to hold the cursor steady without losing visible overlap between the cursor and target (target hold requirement randomly selected for each trial: 50–100 ms, monkey A; 50 ms, monkey C). A limit was placed on the time between target onset and target acquisition (1.5–2 s, monkey A; 1.2–2 s, monkey C). A trial was deemed failed and terminated if visible overlap between cursor and target was lost before satisfying either hold requirement. If all requirements were met, a trial was deemed successful, and the subject was provided with a water reward (120 $\mu$l, monkey A; 120–130 $\mu$l, monkey C). Arms were restrained, and little to no hand movements were observed (although hand positions were not recorded).

The analyzed data were subsets of data from larger experiments. The experimental details for monkey A have been described previously (all no invisible zone conditions from *Chase et al., 2012*). Briefly, each experiment began with roughly 40 trials that were used to calibrate the intuitive BMI mapping (see 'Calibration of the BMI mapping'). Following calibration was a block of $169 \pm 8.1$ successful trials under this intuitive BMI mapping. Next, the BMI mapping was systematically perturbed and held constant for $365 \pm 126$ successful trials. Each perturbation effectively rotated a random subset of recorded units' decoded pushing directions (DPDs), as in *Figure 3—figure supplement 2B*, by a particular angle (5 experiments with 25% of units' DPDs rotated $90°$; 20 experiments with 50% of units' DPDs rotated 60°; 11 experiments with 100% of units' DPDs rotated 30°). In 33 of 36 experiments, perturbation trials were followed by $360 \pm 237$ successful washout trials, during which the perturbation was removed, and the BMI mapping was restored to the intuitive mapping. Unless noted otherwise, analyses of monkey A data refer to intuitive trials. Data from the perturbation and washout trials appear only in *Figure 6*. Each of the 36 experiments comprising these data took place on a unique day.

For monkey C, BMI cursor control alternated between the 2D task (described above) and a 3D task (described below). All monkey C trials analyzed in this work came from the 2D task. Each day began with roughly 40–50 trials to calibrate an intuitive BMI mapping. Following calibration, subsequent blocks alternated between the 2D task and the 3D task, with the first of these tasks chosen randomly each day. The 3D task was similar to the 2D task, except that the cursor was allowed to move in 3D, and targets were distributed about the surface of a workspace-centered sphere. Blocks with the 2D task consisted of $277 \pm 70.4$ trials, and blocks with the 3D task consisted of $527 \pm 252$

trials. Each day consisted of either 3 or 4 blocks. Monkey C experiments did not include trials under a perturbed BMI mapping. The 18 2D blocks analyzed in this work took place on 12 unique days.

## The BMI mapping

BMI cursor position and velocity were determined from recorded spike counts according to a BMI mapping:

$$\mathbf{p}_t = \mathbf{p}_{t-1} + \mathbf{v}_{t-1}\Delta \tag{5}$$

$$\mathbf{v}_t = \mathbf{B}_v\mathbf{u}_t + \mathbf{b}_v \tag{6}$$

where at timestep $t$, $\mathbf{p}_t \in \mathbb{R}^2$ is the cursor position, $\mathbf{v}_t \in \mathbb{R}^2$ is the cursor velocity, $\Delta = 33$ ms is the timestep duration, $\mathbf{u}_t \in \mathbb{R}^q$ is the mean spike count vector recorded simultaneously across $q$ neuronal units over the past 5 timesteps (167 ms), and $\mathbf{B}_v$ and $\mathbf{b}_v$ are the parameters that map neural activity to cursor velocity. Note that the BMI mapping (*Equations 5* and *6*) can be written equivalently in the form of *Equation 1*:

$$\mathbf{x}_t = \mathbf{A}\mathbf{x}_{t-1} + \mathbf{B}\mathbf{u}_t + \mathbf{b} =$$
$$\begin{bmatrix}\mathbf{p}_t \\ \mathbf{v}_t\end{bmatrix} = \begin{bmatrix}\mathbf{I} & \mathbf{I}\cdot\Delta \\ \mathbf{0} & \mathbf{0}\end{bmatrix}\begin{bmatrix}\mathbf{p}_{t-1} \\ \mathbf{v}_{t-1}\end{bmatrix} + \begin{bmatrix}\mathbf{0} \\ \mathbf{B}_v\end{bmatrix}\mathbf{u}_t + \begin{bmatrix}\mathbf{0} \\ \mathbf{b}_v\end{bmatrix} \tag{7}$$

where the cursor state, $\mathbf{x}_t$, concatenates cursor position and velocity.

In some of the following analyses, we required more precise time resolution than could be achieved by analyzing the 5-timestep smoothed velocity commands that drove the BMI cursor (*Equation 6*). For fine-timescale analyses, we defined single-timestep (i.e., unsmoothed) velocity commands as:

$$\mathbf{v}_t^{raw} = \mathbf{B}_v\mathbf{u}_t^{raw} + \mathbf{b}_v \tag{8}$$

where $\mathbf{u}_t^{raw}$ is the vector of recorded spike counts during the single timestep $t$, and $\mathbf{B}_v$ and $\mathbf{b}_v$ are the decoding parameters that were applied online, as in *Equation 6*. Note that $\mathbf{v}_t$ in *Equation 6* is the average of single-timestep velocity commands, $\mathbf{v}_{t-4}^{raw}, \ldots, \mathbf{v}_t^{raw}$.

## Calibration of the BMI mapping

Calibration of parameters $\mathbf{B}_v$ and $\mathbf{b}_v$ of the intuitive BMI mapping was done in closed-loop and followed the population vector algorithm (*Georgopoulos et al., 1983*). Details on this closed-loop calibration have been published previously in *Chase et al. (2012)*. For monkey A, an initial sequence of 8 evenly-spaced radial targets was presented to the subject while the cursor remained stationary at the workspace center. Then, an initial set of BMI parameters was determined by regressing the average spike rates for each trial in this sequence against the corresponding target directions. A second sequence of 8 trials followed, with cursor movements determined by the initial parameter set, but with assistance provided by attenuating velocities perpendicular to target directions. Following this second sequence of trials, new decoding parameters were determined by regressing spike rates from all previous trials against the corresponding target directions. This process was repeated for typically 5 sequences (40 trials), with less assistance in each subsequent sequence until no assistance was provided. The schedule of assistance was determined on an ad-hoc basis. The intuitive BMI mapping calibrated from these trials was then used for the subsequent block of analyzed trials (see 'Behavioral task').

For monkey C, the first task of each day was randomly selected between the 2D and 3D tasks. If the first task was 2D, calibration followed the same procedure as with monkey A. If the first task was 3D, each calibration sequence consisted of 10 targets equidistant from the workspace center. Eight of these targets were on the corners of a workspace-centered cube. The remaining 2 targets were nearly straight out and straight in along the z-direction, but slightly offset so that the cursor was not visually obscured at the central start position. Because these target directions were specified in 3D, calibration regressions resulted in parameters $\mathbf{B}_v^{3D} \in \mathbb{R}^{3\times q}$ and $\mathbf{b}_v^{3D} \in \mathbb{R}^3$ that could map neural activity to 3D velocity. When the task switched to 2D, the parameters $\mathbf{B}_v$ and $\mathbf{b}_v$ were set to the first two rows of $\mathbf{B}_v^{3D}$ and $\mathbf{b}_v^{3D}$, respectively, corresponding to mapped velocities in the frontoparallel plane only. These 3D calibrations typically spanned five 10-trial sequences (50 trials).

## Characterizing inherent visuomotor latencies

BMI subjects experience an inherent visual feedback delay. To assess the visuomotor latency experienced by a subject in our BMI system, we measured the elapsed time between target onset and the appearance of target-related activity in the recorded neural population (*Figure 2A*). To determine the first timestep at which neural activity contained target information, we found the first significant decrease in angular error relative to baseline error. For each trial, baseline error was defined to be the average of absolute angular errors prior to target onset. Here, the angular error at timestep $t$ was defined to be the angle by which the cursor would have missed the target had it continued from its current position, $\mathbf{p}_t$, in the direction of the single-timestep velocity command, $\mathbf{v}_t^{\text{raw}}$, from *Equation 8*. Single-timestep commands ($\mathbf{v}_t^{\text{raw}}$) were analyzed here (as opposed to smoothed cursor velocities, $\mathbf{v}_t$) for improved temporal resolution. Because absolute angular errors range from $0 - 180°$, one might reasonably expect baseline error to be roughly $90°$. Baseline errors shown are less than $90°$ because angular errors were computed relative to the cursor-target overlap zone (i.e., taking into account cursor and target radii; see *Figure 2—figure supplement 1*). When errors were instead computed relative to the target center, baseline errors were roughly $90°$, and identified latencies were unaffected (data not shown). Had we introduced an arbitrary additional delay to the display updates (*Willett et al., 2013*), we would expect a commensurate increase in the identified feedback delay.

## Assessing feedback delay compensation

Because of the visual feedback delay (*Figure 2A*), at timestep $t$ the subject cannot yet directly access the timestep $t$ cursor position. To determine whether subjects compensated for the visual feedback delay, we asked whether neural activity recorded at timestep $t$ was more appropriate for the timestep $t$ cursor position or for a previous cursor position. Across a range of lags, $d = [-100 \text{ ms}, \dots, 300 \text{ ms}]$, we computed the angular errors of single-timestep velocity commands, $\mathbf{v}_t^{\text{raw}}$ (as in *Equation 8*), as if they had originated at lagged positions $\mathbf{p}_{t-d}$ (*Figure 2B*).

Here, angular errors were defined to be the absolute angle by which the cursor would have missed the target had it originated at position $\mathbf{p}_{t-d}$ and continued in the direction of the single-timestep velocity command $\mathbf{v}_t^{\text{raw}}$, taking into account the radii of the cursor and the target (i.e., $\Theta_P$ in *Figure 2—figure supplement 1*). This error metric was chosen because it reflects the task goal, that to succeed in a trial, the subject had to to acquire visible overlap between the cursor and the target (*Figure 2—figure supplement 1*).

By taking into account cursor and target radii, this error metric is influenced by cursor-to-target distance. Specifically, velocity commands originating from positions close to the target will have smaller errors under this definition than the same velocity commands originating far from the target (*Figure 2—figure supplement 1*). Without accounting for this distance-to-target bias, absolute angular errors might appear smaller for lags that are less positive because these lagged cursor positions will tend to be closer to the targets than cursor positions with more positive lags (e.g., $\mathbf{p}_{t-d}$ tends to be closer to the target when $d = 0$ ms than when $d = 300$ ms). To ensure that this distance-to-target bias did not influence our conclusions about feedback delay compensation, errors were computed for the same exact subset of cursor positions across lags. This selection process preserves cursor-to-target distances across lags and thus ensures that the same exact error bias is applied at each lag. To this end, we included in this analysis only cursor positions for which all required lags of neural activity were recorded within the corresponding trial. Further, we only considered cursor positions that were presented at least 100 ms following target onset to ensure that recorded neural activity could plausibly reflect target position given a feedback delay of 100 ms. To determine the error value for a particular lag along the curves in *Figure 2C*, we first averaged all absolute angular errors for that lag within each trial, and then averaged across trials. A preliminary version of this analysis using different experiments has appeared in conference form (*Golub et al., 2012*).

## Framework for internal model estimation (IME)

The IME framework is a statistical tool we developed to extract from neural population activity i) a subject's internal model of the BMI mapping, and ii) the subject's timestep-by-timestep internal predictions about the cursor state. The central concept underlying the IME framework is that at each timestep, the subject internally predicts the current cursor position using outdated visual feedback

and a recollection of previously-issued neural commands (representative of efference copy or corollary discharge [*Crapse and Sommer, 2008*]), and issues the next neural command with the intention of driving the cursor straight toward the target from the up-to-date prediction of the current cursor position (*Figure 3B*).

Formally, the IME framework is a probabilistic model defined by *Equations 9–14* . The subject's internal model, as introduced in *Figure 3A*, is is represented as follows:

$$\text{for } k = \{t-\tau+1,\dots,t\}:$$

$$\tilde{\mathbf{p}}_k^t = \tilde{\mathbf{p}}_{k-1}^t + \tilde{\mathbf{v}}_{k-1}^t \Delta \tag{9}$$

$$\tilde{\mathbf{v}}_k^t = \tilde{\mathbf{A}}_v \tilde{\mathbf{v}}_{k-1}^t + \tilde{\mathbf{B}}_v \mathbf{u}_k^{raw} + \tilde{\mathbf{b}}_v + \mathbf{w}_k^t \tag{10}$$

where $\tilde{\mathbf{p}}_k^t \in \mathbb{R}^2$ and $\tilde{\mathbf{v}}_k^t \in \mathbb{R}^2$ are the subject's internal predictions of the timestep $k$ cursor position and velocity when the subject is sitting at timestep $t$, $\Delta$ is the timestep of the BMI system (33 ms), $\mathbf{u}_k^{\text{raw}} \in \mathbb{R}^q$ is a vector of the spike counts recorded simultaneously across the $q$ neuronal units at timestep $k$, $\tilde{\mathbf{A}}_v \in \mathbb{R}^{2\times2}, \tilde{\mathbf{B}}_v \in \mathbb{R}^{2\times q}$, and $\tilde{\mathbf{b}}_v \in \mathbb{R}^2$ are parameters capturing the subject's internal model, and $\mathbf{w}_k^t \in \mathbb{R}^2$ is a Gaussian random variable (with isotropic noise variance, $w$) representing internal predictions not captured by the internal model. More specifically, $\tilde{\mathbf{A}}$ represents the subject's internal conception of the physical properties of the cursor, and $\tilde{\mathbf{B}}$ represents the subject's internal conception of how neural activity drives movement of the cursor. Note that the subject's internal model in *Equations 9 and 10* can be written in the form of *Equation 2*:

$$\tilde{\mathbf{x}}_k^t = \tilde{\mathbf{A}}\tilde{\mathbf{x}}_{k-1}^t + \tilde{\mathbf{B}}\mathbf{u}_k^{raw} + \tilde{\mathbf{b}} + \text{noise} =$$
$$\begin{bmatrix} \tilde{\mathbf{p}}_k^t \\ \tilde{\mathbf{v}}_k^t \end{bmatrix} = \begin{bmatrix} \mathbf{I} & \mathbf{I}\cdot\Delta \\ 0 & \tilde{\mathbf{A}}_v \end{bmatrix} \begin{bmatrix} \tilde{\mathbf{p}}_{k-1}^t \\ \tilde{\mathbf{v}}_{k-1}^t \end{bmatrix} + \begin{bmatrix} 0 \\ \tilde{\mathbf{B}}_v \end{bmatrix} \mathbf{u}_k^{raw} + \begin{bmatrix} 0 \\ \tilde{\mathbf{b}}_v \end{bmatrix} + \begin{bmatrix} 0 \\ \mathbf{w}_k^t \end{bmatrix} \tag{11}$$

where the subject's internal state prediction, $\tilde{\mathbf{x}}_t^k$, includes the internal prediction of cursor position, $\tilde{\mathbf{p}}_k^t$, and velocity, $\tilde{\mathbf{v}}_k^t$. For simplicity in *Equation 2*, we omitted the noise term, the superscript notation, and the distinction between spike count vectors recorded at a single timestep, $\mathbf{u}_t^{\text{raw}}$, and average spike count vectors across 5 timesteps, $\mathbf{u}_t$ (more details on smoothing are given below).

Visual feedback grounds the subject's internal predictions with reality. At timestep $t$, the subject's internal prediction of the cursor position and velocity at the feedback delay ($\tau$, as discussed in Parameter fitting for the IME framework) match the most recently available cursor position and velocity from visual feedback:

$$\tilde{\mathbf{p}}_{t-\tau}^t = \mathbf{p}_{t-\tau} \tag{12}$$

$$\tilde{\mathbf{v}}_{t-\tau}^t = \mathbf{v}_{t-\tau} \tag{13}$$

The internal model in *Equations 9 and 10* is then applied recursively (i.e., across $k \in \{t-\tau+1,\dots,t\}$) to arrive at up-to-date predictions, $\tilde{\mathbf{p}}_t^t$ and $\tilde{\mathbf{v}}_t^t$, about the current cursor state. The resulting set of internal predictions corresponds to the whiskers shown in *Figure 3*, *Figure 4*, and *Figure 4—figure supplement 1*.

Finally, we incorporate the notion of straight-to-target aiming intention with:

$$\mathbf{G}_t = \tilde{\mathbf{p}}_t^t + \alpha_t \tilde{\mathbf{v}}_t^t + \mathbf{r}_t \tag{14}$$

where $\mathbf{G}_t \in \mathbb{R}^2$ is the target position, $\alpha_t \in \mathbb{R}^+$ is a non-negative distance scale parameter, and $\mathbf{r}_t \in \mathbb{R}^2$ is a Gaussian random variable (with isotropic noise variance, $r$) representing internal velocity predictions that do not point straight to the target. Since the target was held constant within each BMI trial, $\mathbf{G}_t$ took on the same value for all timesteps corresponding to a particular trial. Intuitively, *Equation 14* says that when the subject internally believed the cursor to be at position $\tilde{\mathbf{p}}_t^t$, the intended velocity command, $\tilde{\mathbf{v}}_t^t$, ought to point in the direction of the target, $\mathbf{G}_t$. The distance scale parameters, $\alpha_t$, allow the data to determine the intended speed (i.e., velocity magnitude) at each timestep. This parameterization allows us to avoid imposing a-priori assumptions about the subject's intended speed. During model fitting, larger values of $\alpha_t$ tend to be learned for timesteps when the distance to target is large (from $\tilde{\mathbf{p}}_t^t$), and smaller values tend to be learned when this distance is small. In this

manner, there are no assumptions imposed upon intended speed during model fitting. Rather, the learned internal model determines intended speed from the data. Additionally, the linear form of *Equation 14* was chosen so all latent variables, $\{\tilde{\mathbf{p}}, \tilde{\mathbf{v}}\}$, and observed variables, $\{\mathbf{G}, \mathbf{u}^{\mathrm{raw}}\}$, are jointly Gaussian.

Throughout control, new visual feedback continues to arrive, and new neural commands are issued at each timestep. IME captures this progression by including a new set of internal predictions (i.e., a new whisker) at each timestep. For example, at timestep $t+1$, the subject receives new feedback about the cursor state, $\mathbf{p}_{t-\tau+1}$ and $\mathbf{v}_{t-\tau+1}$, and accordingly forms a new set of internal predictions $\{\tilde{\mathbf{p}}_k^{t+1}, \tilde{\mathbf{v}}_k^{t+1}\}$ for $k \in \{t-\tau+2, \ldots, t+1\}$. The full IME probabilistic graphical model is drawn in *Figure 3—figure supplement 1* to visually depict this instantiation of *Equations 9–14* at each timestep during control.

Through *Equation 14* we assume that the subject attempts to move the cursor straight to the target from an internal estimate of the current position. We believe that straight-to-target aiming is a reasonable first-order assumption because the BMI cursor, on average, moves straight to the target during proficient control (see *Figure 1B*). It may be possible to incorporate other movement objectives, such as minimizing endpoint error (*Harris and Wolpert, 1998*) or movement jerk (*Flash and Hogan, 1985*), in the IME framework, which may yield even greater explanatory power. However, at present, there is not clear evidence that these other movement objectives underlie BMI cursor control, so we apply only the basic straight-to-target movement objective in this work.

Both the BMI mapping (*Equations 5–7*) and the internal model representation (*Equations 9–11*) implement smoothness across BMI cursor velocities and internal velocity predictions, respectively. The details of this smoothing are subtly different between the BMI and the IME framework. To mitigate the effects of neural spiking noise, the BMI mapping smooths cursor velocities by incorporating neural activity at each timestep through the 5-timestep boxcar filter, as described following *Equation 6*. Temporal smoothing in internal velocity predictions is achieved through the subject's internal prior belief about how the internal velocity prediction at one timestep influences the prediction at the next timestep, as encoded by $\tilde{\mathbf{A}}_v$.

In a preliminary IME formulation we presented recently in conference form (*Golub et al., 2013*), the subject's internal state prediction was modeled using position only, rather than using both position and velocity, as we have here. The inclusion of velocity has several important advantages. First, it allows the model to capture the subject using feedback about cursor velocity to internally predict cursor position and velocities. Second, including velocity in the state enables IME to automatically determine the degree of smoothness in internal velocity predictions, based on the data, by fitting an appropriate $\tilde{\mathbf{A}}_v$.

## Parameter fitting for the IME framework

We fit IME models using expectation maximization (EM) (*Dempster et al., 1977*), a maximum likelihood estimation technique for latent variable models. Training data for each trial consisted of recorded spike counts and actual cursor positions for timesteps beginning at movement onset and ending at target acquisition, as well as the target position for that trial. Movement onset for a given trial was defined as the first timestep at which the cursor speed, projected in the center-to-target direction, exceeded 15% of its maximum from that trial. During the E-step, posterior distributions, $P(\{\tilde{\mathbf{x}}\}|\{\mathbf{x}, \mathbf{u}^{\mathrm{raw}}, \mathbf{G}\})$, are computed over the internal states given a set of model parameters. Intuitively, these posteriors are distributions over whiskers that compromise between satisfying the internal model (*Equations 9 and 10*) and straight-to-target aiming (*Equation 14*). During the M-step, these posterior distributions are used to update the model parameters, $\tilde{\mathbf{A}}, \tilde{\mathbf{B}}, \tilde{\mathbf{b}}, w, \{\alpha_t\}$, and $r$. We typically ran EM for 5000 iterations, but allowed fewer iterations if model parameters converged sooner. Although the feedback delay parameter, $\tau$, can be determined using standard model selection techniques (*Golub et al., 2013*), we fixed this parameter ($\tau = 3$, corresponding to 100 ms, monkey A; $\tau = 3$, corresponding to 133 ms, monkey C) for simplicity and to remain consistent with our experimental characterization of the visuomotor latency from *Figure 2A*.

## Computing cross-validated internal model predictions

Throughout our results, if an internal state prediction (whisker) points toward the target, it is not trivially due to our inclusion of straight-to-target aiming into IME (*Equation 14*). Rather, whiskers that point toward targets are evidence of real structure in the data. We ensure that whiskers do not trivially point toward targets by using cross-validation techniques whenever evaluating or visualizing extracted internal models and their corresponding internal state predictions (whiskers). For a given experimental session, trials were randomly assigned to folds such that each fold consisted of one trial to each unique target. We employed $K$-fold cross-validation, where $K$ was the number of folds in a given experimental session. Internal models were fit to the data in $K-1$ folds (training data), and the data from the held-out fold (test data) were used when evaluating the extracted internal model.

Although target positions were used to incorporate the notion of straight-to-target aiming during model fitting (through *Equation 14*), neither targets nor *Equation 14* were used when evaluating extracted internal models on held-out data (relevant for *Figures 3–6*, *Figure 3—figure supplement 3*, *Figure 3—figure supplement 4*, *Figure 3—figure supplement 7*, *Figure 4—figure supplement 1*, and *Figure 4—figure supplement 2*). Rather, whiskers were defined as the expected value of the internal state predictions given only available visual feedback and previously issued neural activity according to the probabilistic model, using only *Equations 9–13* and *not Equation 14*):

$$E\left(\begin{bmatrix} \tilde{\mathbf{x}}_{t-\tau}^{t} \\ \tilde{\mathbf{x}}_{t-\tau+1}^{t} \\ \vdots \\ \tilde{\mathbf{x}}_{t}^{t} \end{bmatrix} | \mathbf{x}_{t-\tau}, \mathbf{u}_{t-\tau+1}^{\text{raw}}, \ldots, \mathbf{u}_{t}^{\text{raw}}\right) = \begin{bmatrix} \mathbf{x}_{t-\tau} \\ \tilde{\mathbf{A}}\mathbf{x}_{t-\tau} + \tilde{\mathbf{B}}\mathbf{u}_{t-\tau+1}^{\text{raw}} + \tilde{\mathbf{b}} \\ \vdots \\ \tilde{\mathbf{A}}\tilde{\mathbf{x}}_{t-1}^{t} + \tilde{\mathbf{B}}\mathbf{u}_{t}^{\text{raw}} + \tilde{\mathbf{b}} \end{bmatrix} \quad (15)$$

We found that cross-validated whiskers consistently pointed straight to targets. This result did not trivially need to be the case, as those targets were not used to construct the whiskers. Rather, given internal models extracted from the training data, the statistical structure underlying the recorded neural activity in the test data was consistent with aiming straight to targets from internal predictions of cursor position.

## Visualizing an extracted internal model

In *Figure 3—figure supplement 2* we visualize the parameters of an extracted internal model as "pushing vectors", and interpret them relative to the corresponding parameters of the BMI mapping. Because of differences in how temporal smoothing is implemented through the BMI mapping and the internal model, magnitudes of pushing vectors are not directly comparable between the BMI mapping and the internal model. In the BMI mapping, temporal smoothing comes from averaging the neural activity across 5 timesteps, as in *Equation 6*. In the internal model, temporal smoothing comes from the specification that each velocity prediction includes a contribution from the previous velocity prediction through $\tilde{\mathbf{A}}_v$, as in *Equation 10*. To provide visually comparable pushing vectors, we factored out the influence of temporal smoothing by visualizing the pushing vectors from $\mathbf{B}_v$ and $\tilde{\mathbf{B}}_v$ as follows. Pushing vectors in *Figure 3—figure supplement 2A* show how the cursor would have moved given a single smoothed spike count from each unit. Analogously in *Figure 3—figure supplement 2B*, we rescaled the pushing vectors in $\tilde{\mathbf{B}}_v$ by $1/\left(1 - \frac{1}{2}\text{trace}(\tilde{\mathbf{A}}_v)\right)$, approximately normalizing by the fraction of the internal velocity prediction that comes from the previous velocity prediction rather than from the current neural activity. The $\frac{1}{2}\text{trace}(\tilde{\mathbf{A}}_v)$ in the scaling factor gives the average value along the diagonal of the $2 \times 2$ matrix, $\tilde{\mathbf{A}}_v$. This normalization was only required because of the particular manner by which cursor velocities were smoothed during BMI experiments. If we had instead used a Kalman filter as the BMI mapping during experiments, pushing vectors would be directly comparable without normalization.

In *Figure 5—figure supplement 1*, we visually interpret an example spike count vector through the internal model shown in *Figure 3—figure supplement 2*. This example spike count vector contributed to the monkey A "movement" bar in *Figure 5B*, as it was the timestep at which cursor-to-target distance first decreased below 50% of the center-to-target distance. The example spike count vector is from the same session as the BMI mapping and internal model parameters shown in *Figure 3—figure supplement 2*, and the spike count vector is from a held-out trial not used to fit that internal model. *Figure 5—figure supplement 1B,C* reflect rescaled $\tilde{\mathbf{B}}_v$ and $\tilde{\mathbf{b}}_v$, as described above.

## Comparison of motor commands predicted by the internal model to those produced by the BMI mapping

Comparisons of the appropriateness of the recorded neural activity through the BMI mapping versus through extracted internal models are shown as angular errors in *Figure 3B,C*, *Figure 4C*, *Figure 6A*, *Figure 3—figure supplement 3*, *Figure 3—figure supplement 4*, *Figure 3—figure supplement 7*, and *Figure 4—figure supplement 2*. For a particular timestep, $t$, we computed the angular error of the neural activity through the BMI mapping as the absolute angle by which the cursor would have missed the target had it continued from cursor position $\mathbf{p}_t$ in the direction of the cursor velocity, $\mathbf{v}_t$ (i.e., $\Theta_P$ in *Figure 2—figure supplement 1*). Similarly, we computed the angular error of the neural activity through the subject's internal model as the absolute angle by which the cursor would have missed the target had it continued from the subject's internal position prediction, $\tilde{\mathbf{p}}_t^t$, in the direction of the subject's internal velocity prediction, $\tilde{\mathbf{v}}_t^t$. Internal model errors were computed from whiskers that could be constructed given cursor feedback and recorded spike counts beginning at movement onset and through target acquisition. Whiskers were extracted using the cross-validation techniques described in Computing cross-validated internal model predictions.

## Assessing whether internal model mismatch could appear as a spurious result due to correlated spiking variability

An alternative explanation of our data could be that the subject's internal model is well-matched to the BMI mapping, but that correlated noise in neural firing leads us to estimate an internal model that rejects noise better than the BMI mapping. To determine whether our finding of internal model mismatch might have been a spurious result of noise in the recorded neural activity, we performed the following simulation, which assumes the alternative hypothesis that there is no internal model mismatch. First, we simulated neural activity under the assumption that the BMI mapping and the internal model are equal (i.e., the alternative hypothesis). Then, we evaluated that simulated neural activity through the BMI mapping and the extracted internal model (which were not equal). The key insight provided is due to the ability to explicitly define signal versus noise in simulation. Although there are many possible ways to define signal versus noise in the recorded neural activity, here we assume the internal model and the BMI mapping are equal (the alternative hypothesis), and we define signal to be the component of a neural activity pattern that maps to the subject's desired movement direction through that mapping. We define noise to be the residual neural activity pattern after subtracting out the signal.

We began with a set of 32 desired movement directions, $d_i^* \in \{0°, 11.25°, 22.5°, \ldots\}$. This set was chosen to align with the 16 target directions with an additional direction halfway between each pair of adjacent targets. We labeled each recorded neural activity pattern, $\mathbf{u}_t^{\text{raw}}$, according to the direction, $d_i^*$, that it most closely matched after being passed through the BMI mapping (*Equation 8*) from that experiment. This labeling procedure produces, for each direction, $d_i^*$, a set of real recorded neural activity patterns, $\mathscr{U}_i$, that reflect the intention to move in direction $d_i^*$. For each direction, we then defined an idealized neural activity pattern to be the mean of all real neural activity patterns labeled as matching that direction through the BMI mapping:

$$\mathbf{u}_i^* = \frac{1}{|\mathscr{U}_i|} \sum_{\mathbf{u}_t^{\text{raw}} \in \mathscr{U}_i} \mathbf{u}_t^{\text{raw}} \tag{16}$$

where $|\mathscr{U}_i|$ is the number of real activity patterns labeled as matching direction $d_i^*$. We performed this procedure separately for each intuitive session. Idealized neural activity patterns were calculated from sets of $109 \pm 24$ (monkey A) and $178 \pm 56$ (monkey C) real neural activity patterns (mean $\pm$ standard deviation across all experiments and directions). We evaluated the error of these idealized neural activity patterns through the BMI mapping and through the extracted internal model, relative to the corresponding desired direction (i.e., the average absolute angular error between $d_i^*$ and $\mathbf{v}_i^* = \mathbf{B}\mathbf{u}_i^* + \mathbf{b}$, and between $d_i^*$ and $\tilde{\mathbf{v}}_i^* = \tilde{\mathbf{B}}\mathbf{u}_i^* + \tilde{\mathbf{b}}$, respectively) (*Figure 3—figure supplement 6A*). By construction, we expect errors through the BMI mapping to be nearly zero (nonzero errors are due to the discretization of direction).

To determine the effect of noise in the recorded neural activity, we corrupted these idealized neural activity patterns by combining them with simulated noise patterns drawn from residuals in the

recorded neural activity. Residuals, $\mathbf{c}_t$, were computed by subtracting the idealized neural activity pattern, $\mathbf{u}_i^*$, from the recorded neural activity patterns, $\mathbf{u}_t^{\text{raw}}$, corresponding to that idealized pattern:

$$\text{for each} \quad \mathbf{u}_t^{\text{raw}} \in \mathscr{U}_i, \quad \mathbf{c}_t = \mathbf{u}_t^{\text{raw}} - \mathbf{u}_i^* \tag{17}$$

Simulated noise patterns were then drawn from the across-direction set of residuals:

$$\mathbf{s}_k \sim \{\mathbf{c}_t\}_{\forall t} \tag{18}$$

Finally, simulated noisy neural activity patterns, $\mathbf{u}_{i,k}^{\text{sim}}$, were formed by combining the idealized neural activity patterns with the simulated noise patterns:

$$\mathbf{u}_{i,k}^{\text{sim}} = \mathbf{u}_i^* + \mathbf{s}_k \tag{19}$$

We evaluated the error of the simulated noisy neural activity patterns, $\mathbf{u}_{i,k}^{\text{sim}}$, through the BMI mappings and through the extracted internal models, relative to the corresponding desired direction (that is, the average absolute angular error between $d_i^*$ and $\mathbf{v}_{i,k}^{\text{sim}} = \mathbf{B}\mathbf{u}_{i,k}^{\text{sim}} + \mathbf{b}$, and between $d_i^*$ and $\tilde{\mathbf{v}}_{i,k}^{\text{sim}} = \tilde{\mathbf{B}}\mathbf{u}_{i,k}^{\text{sim}} + \tilde{\mathbf{b}}$, respectively) (*Figure 3—figure supplement 6B*). This analysis was fully cross-validated, meaning that we only evaluated a simulated neural activity pattern through an internal model if its simulated noise pattern was not computed from a recorded neural activity pattern used during fitting of that internal model. To further match the statistics of the real data, we ensured that we evaluated the same number of simulated neural activity patterns corresponding to a particular desired direction as the number of recorded neural activity patterns that matched that desired direction through the BMI mapping.

## Evaluating the speed bias resulting from internal model mismatch

In *Figure 5* we compared the timestep-by-timestep speeds of the actual cursor to the subject's intended cursor speed, as determined by extracted internal models. At timestep $t$, actual cursor speed was taken to be the magnitude of cursor velocity $\mathbf{v}_t$ (*Equation 6*), and intended cursor speed was taken to be the magnitude of the subject's velocity belief, $\tilde{\mathbf{v}}_t^t$. To form the curves in *Figure 5A*, we selected all timesteps when intended cursor speed was $s$ and computed the distribution of actual cursor speeds at those same timesteps. Curves show the mean actual cursor speed (and S.E.M.) as a function of intended cursor speed. In *Figure 5B*, we included all timesteps preceding target onset to form the speed difference bars labeled "center hold." To form the "movement" bars, we included for each trial the single timestep at which cursor-to-target distance first decreased below 50% of the center-to-target distance.

## Visualizing temporal dynamics of neural activity and internal models during adaptation to perturbations

In *Figure 6A* we interpreted monkey A neural activity through several relevant mappings: the intuitive BMI mapping, the perturbed BMI mapping, the time-varying internal model, and the late intuitive internal model. The time-varying internal model was extracted from a moving window of 48 trials and was updated every 16 trials (1 trial to each of the 16 targets). The late intuitive internal model was extracted from the last 48 trials during the intuitive session. For each mapping, whiskers were constructed at each timestep and angular errors were evaluated relative to the target perimeter. In general, these errors describe how task-appropriate the subject's neural activity was for a particular mapping at a particular moment during the experiments. In the case of the BMI mappings, each whisker equates to how the cursor position and velocity would have evolved from a particular position on the actual cursor trajectory (i.e., the visual feedback) had that BMI mapping been in effect. In the case of the intuitive BMI mapping during the intuitive and washout trials and in the case of the perturbed BMI mapping during the perturbation trials, these whiskers by definition exactly match the cursor trajectories displayed during the experiments. When a particular BMI mapping was not in effect (e.g., the intuitive BMI mapping during the perturbation trials), these whiskers describe how the cursor would have moved under that BMI mapping and thus would not match the cursor behavior from the experiments.

In *Figure 6B* we evaluated the differences between the time-varying internal model and the BMI mappings. We interpreted monkey A neural activity through the BMI mappings and the time-varying

internal model, and at each timestep, computed the angle between the velocity predicted by the internal model (i.e., by constructing a whisker) and the velocity computed through each BMI mapping. For experiments with a counter-clockwise rotational component to the perturbation, signs of angles were flipped so that nonzero biases would not cancel out when averaging across experiments. Angles were computed after centering the origins of the two velocity vectors. In contrast to the absolute angles computed relative to the target, which are presented through this work, these angles were signed and computed without using the target position. Analyzing signed angles permits quantification of the bias in these angles, whereas absolute angles are better suited for quantifying the variance of errors. During the intuitive trials, angular errors through the subject's internal model and through the intuitive BMI mapping tend to be unbiased (i.e., the average signed angular error is roughly zero for both; data not shown) but with different variances (hence the substantial differences in absolute angular errors in *Figure 3C* and *Figure 6A*). Angular errors relative to the target become biased during the perturbation and washout trials due to the rotational nature of the perturbations. We also analyzed the signed angular errors (data not shown) to ensure that these biases were not affecting our interpretation of the errors in *Figure 6A*.

The neural activity used in this time-varying difference metric was taken from 16-trial non-overlapping blocks of trials aligned to the horizontal axis labels (as in *Figure 6A*). Results were nearly identical when instead using a fixed set of neural activity taken from the first 48 trials of the intuitive session (data not shown). We also evaluated a number of additional difference metrics, including angles between unsmoothed velocities (i.e., between $(\tilde{\mathbf{B}}_v\mathbf{u}_t^{\mathrm{raw}} + \tilde{\mathbf{b}}_v)$ and $(\mathbf{B}\mathbf{u}_t^{\mathrm{raw}} + \mathbf{b})$; see *Equation 10*), cartesian distances between whisker endpoints (i.e., $||\tilde{\mathbf{p}}_t^t - \mathbf{p}_t||_2$), and angles between the corresponding columns of the $\mathbf{B}$ and $\tilde{\mathbf{B}}$ matrices from the BMI mappings and internal models, respectively (data not shown). These metrics were consistent with the metric of *Figure 6B* in showing that the time-varying internal model becomes more similar to the particular perturbed BMI mapping during the perturbation trials and to the particular intuitive BMI mapping during the washout trials.

## Acknowledgements

We thank AB Schwartz and AS Whitford (Motorlab, University of Pittsburgh) for access to BCI data. We thank AP Batista, O Donchin, LL Holt, CR Olson, and MA Smith for critical discussions and reading of this manuscript. This work was supported by NSF IGERT Fellowship (MDG), NIH-NICHD-CRCNS-R01-HD-071686 (BMY), and PA Department of Health Research Formula Grant SAP#4100057653 under the Commonwealth Universal Research Enhancement program (SMC).

## Additional information

### Funding

| Funder | Grant reference number | Author |
|---|---|---|
| National Science Foundation | IGERT | Matthew D Golub |
| National Institutes of Health | NIH-NICHD-CRCNS-R01-HD-071686 | Byron M Yu |
| Pennsylvania Department of Health | C.U.R.E. SAP#4100057653 | Steven M Chase |

The funders had no role in study design, data collection and interpretation, or the decision to submit the work for publication.

### Author contributions

MDG, BMY, Conception and design, Analysis and interpretation of data, Drafting or revising the article; SMC, Conception and design, Acquisition of data, Analysis and interpretation of data, Drafting or revising the article, Contributed unpublished essential data or reagents

### Author ORCIDs

Matthew D Golub, http://orcid.org/0000-0003-4508-0537

## Ethics

Animal experimentation: All animal procedures were approved by the Institutional Animal Care and Use Committee (IACUC) of the University of Pittsburgh (protocol 0808279).

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
