## [Decision Letter]

Thank you for submitting your work entitled "Internal models for interpreting neural population activity during sensorimotor control" for peer review at *eLife*. Your submission has been favorably evaluated by Timothy Behrens (Senior editor), and two reviewers.

The reviewers have discussed the reviews with one another and the Reviewing editor has drafted this decision to help you prepare a revised submission.

Summary:

This paper describes a novel statistical analysis of population neural activity in M1 during brain-machine interface (BMI) control. It reports that errors in BMI control can be understood in terms of a mismatch between a true control model and an "internal" control model assumed by the animal. It shows that population activity accurately takes account of the time-lag in visual feedback, and compensates for a perturbation of the true model by a corresponding perturbation of the internal model.

The paper is well written, interesting and highly original, and is likely to be of broad interest to the BMI community as well as to the broader community of neuroscientists interested in issues of population coding, dimensionality reduction, and motor control.

Essential revisions:

1) First, I admit I am struggling a little bit to understand why I shouldn't adopt the conclusion offered in the Discussion that "it might be tempting to view an extracted internal model as simply the BMI mapping that should have been identified during calibration." The authors give several arguments here but sometimes I don't entirely follow them. The true model here is linear difference equation. The (fitted) internal model also takes the form of a linear difference equation, and accounts for a substantial fraction of the errors. This indicates (if I understand the claims correctly) that if the experimenters had used the learned internal model instead of the "true" model to drive cursor position, the BMI performance would have been much better. So I'm puzzled about why the authors don't feel that this is the straightforward conclusion of the paper. (Couldn't this be considered a positive result, i.e., a finding that suggests a much better way to improve BMI training? I suppose a drawback to this framing is that if the paper were pitched as a better way to train BMIs, we would presumably expect to see that it actually works, i.e., show some data where the estimated internal model is used to drive behavior and achieves much lower error).

2) A related concern: the paper asserts that the good performance of the estimated internal model shows that the limits to performance are the observer's inaccurate estimates of the model, as opposed to noise in the neural activity. I'd like to suggest a third possibility: could it possible that it's not *noise*, but rather systematic problems in producing neural activity patterns needed to optimally drive the BMI? For example, suppose the animal could only produce activity patterns corresponding to motion in the 4 cardinal directions. Then it would to alternate between producing these patterns in order to drive a diagonal cursor movement. Or (more realistically), suppose there were only certain fixed points in neural activity space that M1 can easily visit. These fixed points might be systematically misaligned with the directions needed to optimally drive the controller. Re-estimating the internal model allows the best linear mapping from these allowed neural activity patterns to those that can drive the cursor as needed during the experiment.

I raise this possibility in part because it seems like this was the conclusion of the Yu & Batista groups' nature paper last year, showing that control within an intrinsic manifold (as defined by the state space explored by the network prior to BMI activity) is much easier than outside this manifold. I realize it is likely that the "intuitive" training procedure would latch on to "out of manifold" patterns, since it tries to use activity during training to define the controller. But it would still be nice to rule out this kind of effect – could it be that rather than insufficient knowledge of the controller, the difference results from differences in the ease with which certain network states can be visited?

3) Result one is that compensation for feedback delay indicates the existence of an internal model. My confusion is that, because equation 1 includes cursor velocity as an input, couldn't it be doing this extrapolation all by itself? How does extrapolation provide evidence of an internal model if the external model could be doing it? Don't you need to show that the internal model is doing a better job of extrapolation that equation 1 possibly could?

4) The second issue is whether it is justified to call equation 2 an approximation of an internal model. The authors discuss reasons why equation 1 and 2 are not identical, that is, why the internal and external models don't match. Let me provide a simplified analogy. Suppose a single scalar were being extracted from u by x = Bu (in other words, ignore A and b for the moment). In the initial "intuitive BMI mapping" B is defined by a procedure that is not well explained. Suppose that, on each trial u = u_0_ + s, where u_0_ would give perfect performance and s is noise. Clearly Bu_0_ = x (the correct x), but has B been chosen to minimize the effects of noise? That is, has Bs been minimized? If the noise is correlated, this can be done. What if B~ is simply more orthogonal to the major PCs of s than B? Wouldn't this explain the effect? Even if the parameters of equation 1 are optimal with respect to noise, how do we know that the correlation structure of s has not changed by the time the model of equation 2 was constructed?

In summary, the paper would be strengthened by a more convincing case that evidence for an internal model has been obtained and that equation 2 can be interpreted as an approximation for this internal model.

---

## [Author Response]

*Essential revisions:*

*1) First, I admit I am struggling a little bit to understand why I shouldn't adopt the conclusion offered in the Discussion that "it might be tempting to view an extracted internal model as simply the BMI mapping that should have been identified during calibration." The authors give several arguments here but sometimes I don't entirely follow them. The true model here is linear difference equation. The (fitted) internal model also takes the form of a linear difference equation, and accounts for a substantial fraction of the errors. This indicates (if I understand the claims correctly) that if the experimenters had used the learned internal model instead of the "true" model to drive cursor position, the BMI performance would have been much better. So I'm puzzled about why the authors don't feel that this is the straightforward conclusion of the paper. (Couldn't this be considered a positive result, i.e., a finding that suggests a much better way to improve BMI training? I suppose a drawback to this framing is that if the paper were pitched as a better way to train BMIs, we would presumably expect to see that it actually works, i.e., show some data where the estimated internal model is used to drive behavior and achieves much lower error).*

We agree with the reviewers. Our results do suggest that BMI performance might be improved by applying the learned internal model as the BMI mapping. We believe this to be an exciting direction for future experiments and future publications. We have decided to cut the paragraph in question, (“it might be tempting to view an extracted internal model as simply the BMI mapping that should have been identified during calibration”), which was originally intended to steer readers away from misinterpreting our study as an (incremental) engineering advance, rather than a novel approach to the basic scientific study of sensorimotor control. The preceding paragraph covers this sufficiently, and the following paragraph highlights the potential future direction of applying an extracted internal model during closedloop control.

The section now reads as follows:

“An extracted internal model and a BMI mapping are closely related. They take a similar mathematical form (Equations 12) and both project highdimensional population activity to a lowdimensional kinematic space. […] Future studies will be required to determine whether further improvements in performance might be possible by using the IME framework toward designing the BMI mapping.”

*2) A related concern: the paper asserts that the good performance of the estimated internal model shows that the limits to performance are the observer's inaccurate estimates of the model, as opposed to noise in the neural activity. I'd like to suggest a third possibility: could it possible that it's not noise, but rather systematic problems in producing neural activity patterns needed to optimally drive the BMI? For example, suppose the animal could only produce activity patterns corresponding to motion in the 4 cardinal directions. Then it would to alternate between producing these patterns in order to drive a diagonal cursor movement. Or (more realistically), suppose there were only certain fixed points in neural activity space that M1 can easily visit. These fixed points might be systematically misaligned with the directions needed to optimally drive the controller. Re-estimating the internal model allows the best linear mapping from these allowed neural activity patterns to those that can drive the cursor as needed during the experiment.*

*I raise this possibility in part because it seems like this was the conclusion of the Yu & Batista groups' nature paper last year, showing that control within an intrinsic manifold (as defined by the state space explored by the network prior to BMI activity) is much easier than outside this manifold. I realize it is likely that the "intuitive" training procedure would latch on to "out of manifold" patterns, since it tries to use activity during training to define the controller. But it would still be nice to rule out this kind of effect – could it be that rather than insufficient knowledge of the controller, the difference results from differences in the ease with which certain network states can be visited?*

This is an important concern, and we thank the reviewers for bringing this up. To address this, we included new analyses (Figure 3—figure supplement 5), which show that the distribution of observed lowdimensional mapped cursor velocities covers the full range of directions. This suggests that our main finding cannot be explained by subjects’ inability to produce particular velocities through the BMI mapping. We also made the following textual changes. We now mention this possibility when introducing our central hypothesis early in the Results section:

“The central hypothesis in this study is that movement errors arise from a mismatch between the subject’s internal model of the BMI and the actual BMI mapping. […] subjects’ inability to produce certain patterns of neural activity, or subjects disengaging from the task.”

Then, we fully address the possibility in a new subsection of the Results section, entitled “Two alternative hypotheses do not explain the internal model mismatch effect.”

“The data presented thus far support our central hypothesis that internal model mismatch is a primary source of movement errors. […] This suggests that our main finding of internal model mismatch cannot be explained by subjects’ inability to produce particular neural activity patterns.”

*3) Result one is that compensation for feedback delay indicates the existence of an internal model. My confusion is that, because equation 1 includes cursor velocity as an input, couldn't it be doing this extrapolation all by itself? How does extrapolation provide evidence of an internal model if the external model could be doing it?*

We realized that our description of this analysis in the manuscript may have lead to some confusion. The reviewers are correct that equation 1 (the BMI mapping) can be used to perfectly extrapolate the cursor trajectory (given all recorded neural activity as inputs). The question, however, is whether the subject does something like this to internallypredict the path of the cursor over the ~100ms visual feedback latency. The idea behind our feedback compensation result (Figure 2) is that the subjects’ neural activity reflected the direction from the current cursor position to the target more so than from any previously displayed cursor position. This suggests an internal model being used by the subject to internally predict cursor motion. To address this potential source of confusion, we have added the following paragraph to the end of the Results section, “Subjects compensate for sensory feedback delays while controlling a BMI”:

“Because we have not yet explicitly identified the subject's *internal* model, motor commands were defined in this analysis using the BMI mapping, which is *external* to the subject. […] we next asked whether we could explicitly identify an internal model from the recorded neural activity.”

*Don't you need to show that the internal model is doing a better job of extrapolation that equation 1 possibly could?*

While equation 1 can perfectly reconstruct the cursor trajectory (since this is what defined the cursor trajectory during the experiment), the subject cannot directly know the form or parameters of the equation. Rather, the subject builds up an internal model, which we represent with equation 2. In Figure 3, we do just as the reviewer suggests – show that the internal model (equation 2) does a better job of explaining the recorded neural activity than does the BMI mapping (equation 1).

*4) The second issue is whether it is justified to call equation 2 an approximation of an internal model. The authors discuss reasons why equation 1 and 2 are not identical, that is, why the internal and external models don't match. Let me provide a simplified analogy. Suppose a single scalar were being extracted from u by x = Bu (in other words, ignore A and b for the moment). In the initial "intuitive BMI mapping" B is defined by a procedure that is not well explained. Suppose that, on each trial u = u_0_ + s, where u_0_ would give perfect performance and s is noise. Clearly Bu_0_ = x (the correct x), but has B been chosen to minimize the effects of noise? That is, has Bs been minimized? If the noise is correlated, this can be done. What if*
B˜
*is simply more orthogonal to the major PCs of s than B? Wouldn't this explain the effect? Even if the parameters of equation 1 are optimal with respect to noise, how do we know that the correlation structure of s has not changed by the time the model of equation 2 was constructed?*

*In summary, the paper would be strengthened by a more convincing case that evidence for an internal model has been obtained and that equation 2 can be interpreted as an approximation for this internal model.*

This is another important concern, and we thank the reviewer for bringing it to our attention. To address this, we conducted a simulation along the lines of the scenario that the reviewer describes. Our simulation, which was carefully designed to match the statistics of the real recorded neural activity, shows that our internal model mismatch result cannot be explained by a scenario with correlated noise and no internal model mismatch. These simulated data are presented in a new Figure 3—figure supplement 6. We added the following text into the Results subsection, entitled “Two alternative hypotheses do not explain the internal model mismatch effect”:

“Second, we explored the possibility that the subject intended to produce neural commands that were correct according to the BMI mapping, but that those intended commands were corrupted by “noise'' that is oriented such that errors appear smaller through the extracted internal model than through the BMI mapping. […] Simulated neural activity was more consistent with the BMI mapping than the extracted internal model, which contrasts with our finding from the recorded neural activity.”

We have included additional details about this new analysis in a new Materials and methods subsection, “Assessing whether internal model mismatch could appear as a spurious result due to correlated spiking variability”: